# Structural exposure of different microtubule binding domains determines the propagation and toxicity of pathogenic tau conformers in Alzheimer's disease

Lenka Hromadkova[1¤a], Chae Kim[1], Tracy Haldiman[1], Mohammad Khursheed Siddiqi[1¤b], Krystyna Surewicz[2], Kiley Urquhart[1], Dur-E-Nayab Sadruddin[1], Lihua Peng[1], Xiongwei Zhu[1], Witold K. Surewicz[2], Mark L. Cohen[1,3], Mark R. Chance[4,5], Rohan de Silva[6], Janna Kiselar[4,5], Jiri G. Safar [1,7,8]*

**1** Department of Pathology, Case Western Reserve University School of Medicine, Cleveland, Ohio, United States of America, **2** Department of Physiology and Biophysics, Case Western Reserve University, Cleveland, Ohio, United States of America, **3** National Prion Disease Pathology Surveillance Center, Case Western Reserve University School of Medicine, Cleveland, Ohio, United States of America, **4** Center for Proteomics and Bioinformatics, Case Western Reserve University School of Medicine, Cleveland, Ohio, United States of America, **5** Department of Nutrition, Case Western Reserve University School of Medicine, Cleveland, Ohio, United States of America, **6** Reta Lila Weston Institute of Neurological Studies and Department of Molecular Neuroscience, UCL Institute of Neurology, London, United Kingdom, **7** Departments of Neurology, Case Western Reserve University School of Medicine, Cleveland, Ohio, United States of America, **8** Departments of Neurosciences, Case Western Reserve University School of Medicine, Cleveland, Ohio, United States of America

¤a Czech Centre for Phenogenomics, Institute of Molecular Genetics of the Czech Academy of Sciences, Prague, Czech Republic
¤b Department of Anatomy and Neurobiology, Virginia Commonwealth University, Richmond, Virginia, United States of America
* jiri.safar@case.edu

## Abstract

Deposits of misfolded tau proteins are leading indicators of cognitive decline in Alzheimer's disease (AD), and our recent data implicate distinctly misfolded conformers of the tau protein with high seeding potency in rapid progression. We considered prion-like templated propagation of misfolding in neurons as an underlying mechanism and derived sensitive conformational assays to test this concept and identify critical structural drivers. Using novel photochemical hydroxylation monitored with a panel of Europium-labeled monoclonal antibodies, we investigated the structural organization of different microtubule binding domains (MTBDs) in brain-derived tau conformers in AD with different progression rates. We analyzed the impact of structural organization of different MTBDs on seeding potency *in vitro* and in primary neurons, and on the propagation rate of tau misfolding, compartmentalization, cytotoxicity, and calcium homeostasis in neuronally differentiated SH-SY5Y cells. Within the extensive inter-individual structural variability in all MTBDs and C-terminal tails, the most significant driver of seeding potency and propagation of tau protein misfolding in both *in vitro* seeding assays and in neuronal cultures was the structural

**Data availability statement:** All relevant data are in the manuscript and its supporting information files.

**Funding:** This work was supported by the National Institute on Aging (1RF1AG058267 and 1RF1AG061797 to JGS), Alzheimer's Association (AARF-22-918090 to LH), National Center for Chronic Disease Prevention and Health Promotion (UR8/CCU515004 to MLC), National Institute on Aging (AG072959 to XZ), and by National Institutes of Health (S10-OD024996 to LH). The funders had no role in study design, data collection and analysis, decision to publish, or preparation of the manuscript.

**Competing interests:** The authors have declared that no competing interests exist.

exposure of the fourth MTBD (R4). In contrast, the major driver of calcium influx induced in neurons by the accumulation of misfolded tau was the structural exposure of the R1 domain. The data provide compelling evidence for a major diversity in the structural organization of MTBDs of misfolded AD brain-derived tau protein and implicate the structural exposure of distinct domains in different pathogenetic steps of AD — R4 tau domain in progression rate, and R1 domain in variable synaptic toxicity of misfolded tau, and thus in cognitive decline.

## Author summary

Within the spectrum of Alzheimer's disease (AD) phenotypes, the disease stage, brain areas of atrophy, and ultimate cognitive decline tend to correlate with the progressive spread of misfolded tau protein aggregates, and our recent findings implicated the distinct, highly potent tau seed conformers in the rapid progression of AD. Our working hypothesis was that the specific structural organization of microtubule-binding domains (MTBDs) in pathogenic tau conformers is a critical driver of their replication by controlling their affinity to monomers of normal tau, thus leading to different rates of replication and propagation in neurons. The data presented here support the hypothesis and demonstrate (i) major inter-individual structural diversity of MTBDs in tau conformers isolated from AD with different progression rates, particularly in the fourth repeat (R4) tau domain, (ii) a significant role of the structural organization and exposure of the R4 domain within the MTBDs in the replication and propagation of tau conformers, and (iii) a striking correlation between structural exposure of the first repeat (R1) domain and toxicity in neuronal cultures. Although specifically targeting these domains in prion strain-like tau conformers with inhibitors will be challenging, it is a potentially explorable strategy in future studies.

## Introduction

Alzheimer's disease (AD) demonstrates considerable variation in clinical phenotypes across the AD population. We recently identified a malignant rapidly progressive AD (rpAD) subgroup with a median disease duration of only 8 months, in contrast to 9 years in all late-onset AD cases collected in the National Alzheimer's Coordinating Center (NACC) database [1–7]. Depending on the definition — a drop of 6 or more points per year in Mini-Mental State Examination (MMSE) and/or less than 36-month survival — 10% - 30% of all AD cases are classified as rpAD [1,4,6–8]. Our data and the rpAD cohorts investigated independently in the centers in Germany, Japan, Spain, and France in the past decade demonstrated uniform NIA-AA neuropathological characteristics, low frequency of e4 alleles in the APOE gene, higher frequency of novel loci in AD risk genes while the autosomal dominant history of dementia or comorbidity is absent [3,4,6,7,9–14]. The interactomes of amyloid beta and

hyperphosphorylated tau proteins differ significantly in rpAD from those in slowly progressive cases [13,14]; however, the pathogenetic mechanism of distinct progression rates in AD is not understood [15–18].

The considerable progress in high-resolution structural studies of tau filaments by cryo-EM has confirmed their conformational diversity in AD, corticobasal degeneration (CBD), chronic traumatic encephalopathy (CTE), and some other tauopathies [19–23]. Moreover, our direct conformational data on patients with Frontotemporal Lobar Degeneration (FTLD) with the same MAPT P301L mutation demonstrated three distinct structural signatures of misfolded tau aggregates in different clinical phenotypes, two of which resembled those found in aged mice in the TgTAUP301L model [24]. These findings are supported by a new *in vitro* cryo-EM and NMR study demonstrating the assembly of disease-specific tau filaments via polymorphic intermediates [25], indicating remarkable conformational plasticity of the misfolded tau protein [26]. Consequently, the distinctly misfolded conformers of highly potent four-repeat (4R) tau seeds we found recently in rpAD patients [3,27] raise a series of critical mechanistic questions regarding the role of pathogenic conformers of tau in distinct phenotypes of AD and which specific structural domains (epitopes) of misfolded tau drive the propagation, toxicity, and apparent diversity of pathological and clinical features in AD. Addressing these questions is fundamental for verifying and advancing the emerging concept, pointing to the structurally distinct conformers of tau as critical differentiating factors in AD.

Testing for phenotype-specific conformational differences between human brain-derived tau conformers in fibrillar, prefibrillar (oligomeric), and monomer structures presents enormous challenges because of the lack of adequate experimental approaches suitable for all these major forms. Hydroxyl radical (•OH) protein footprinting is a structural covalent labeling technique that uses •OH generated *in situ* to probe the structure of native proteins regardless of their higher-order arrangement — monomeric, oligomeric, or fibrillar [28–31]. We expanded the footprinting methodology to a new application that exploits the loss of antibody affinity due to the progressive hydroxylation of amino acid side chains within each antibody epitope. Using photochemical hydroxylation and five monoclonal antibodies with a high affinity for linear epitopes covering all four repeat domains of the tau core and C-terminal tail, we investigated the structural organization of different isoforms of the tau protein in 22 AD cases with variable progression rates. Hydroxylation footprinting and parallel monitoring of seeding potency in *in vitro* assays and neuronal cultures have allowed the investigation of the impact of different repeat domains and their structural organization on the propagation of misfolding. Our findings demonstrate the major structural diversity of each MTBD domain and establish a link between the structural organization of a particular domain in the misfolded tau protein conformers and their replication. These data are consistent with the growing body of evidence that patients with AD may have a distinct structural organization of tau epitopes that are critical for propagation and toxicity, emphasizing the need for future studies targeting these domains with specific high-affinity antibodies.

## Results

### Demographics, concentrations, and conformational characteristics of tau proteins in the spectrum of AD phenotypes

To investigate the full spectrum of AD phenotypes and disease durations, AD cases were randomly selected from two different Case Western Reserve University (CWRU) biobanks: the National Prion Disease Pathology Surveillance Center (NPDPSC) and the Brain Health and Memory Center of the Neurological Institute at University Hospitals Case Medical Center in the repository of the Department of Pathology at CWRU [3,4,32] (Methods). The cumulative Kaplan Meier survival curves (Fig 1A), age distribution, neuropathology, APOE allelic frequency, concentrations, and conformational characteristics of Amyloid beta (Table 1) in the cortex were within the ranges we reported previously and documented the clinicopathological heterogeneity of AD [1–3,11,12,33,34]. The conformational stability assay (CSA) was derived from previously validated protocols for differentiating distinct conformers of human and animal prions, amyloid beta, and tau [4,35–45] and data in the frontal cortex (Fig 1B) correspond to the variable conformational stability of AD brain-derived tau aggregates we reported in the hippocampal cortex previously [3]. The concentrations and conformational characteristics

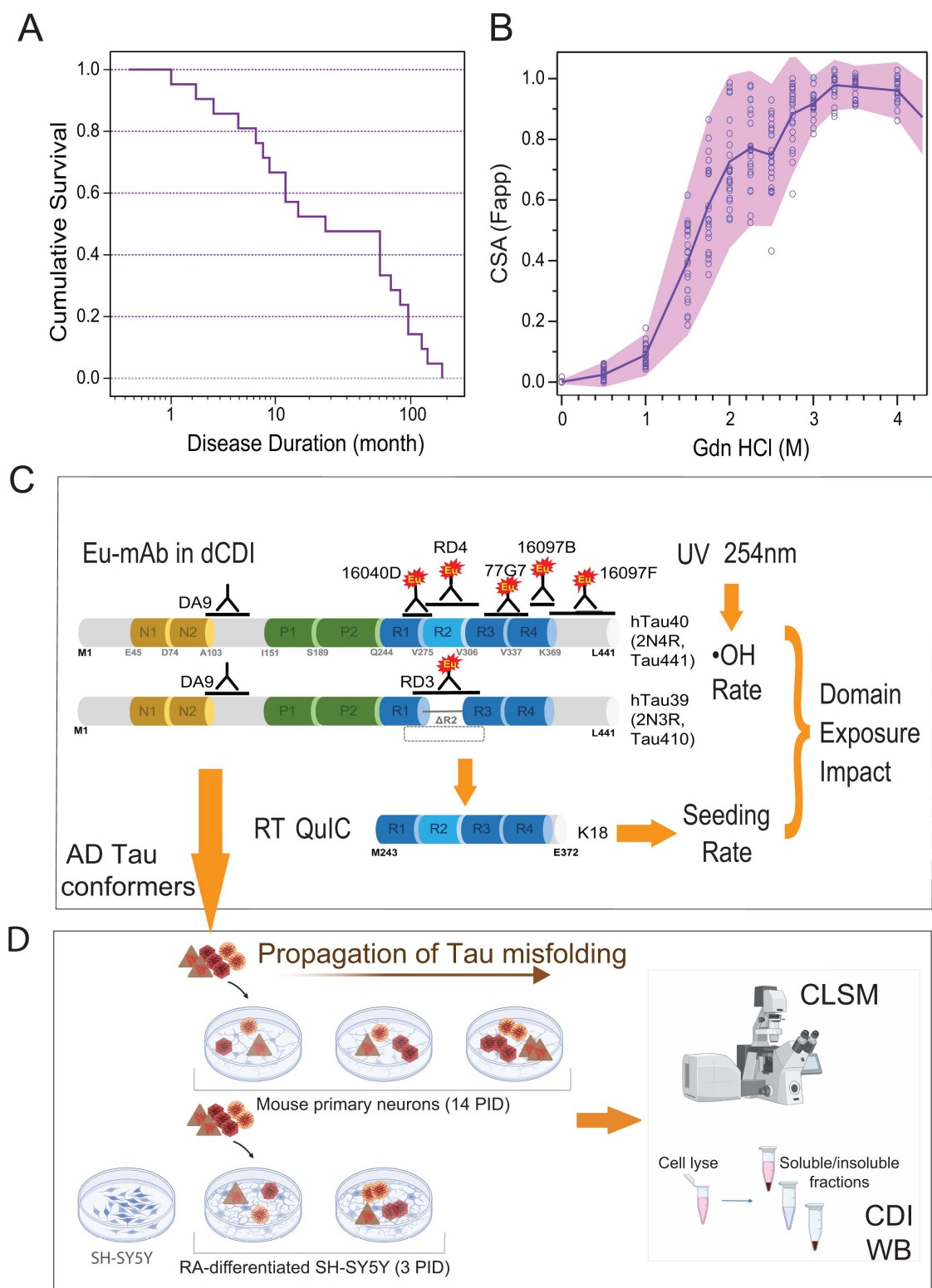

**Fig 1. Progression rates of AD, conformational profiles of insoluble tau protein, and schematic representation of experimental design. (A)** Kaplan-Meier cumulative survival analysis and duration of disease in pathologically verified AD (n = 22). **(B)** Conformational diversity of brain-derived Sarkosyl-insoluble tau in the cortex of AD determined in the brain tissue homogenates (n = 22) with conformational stability assay (CSA) [3]. The line and

shade represent mean ± S.E.M. **(C)** Schematic representation of photochemical hydroxylation method and the antibodies used for monitoring different domains. The tau protein domains are not to scale with microtubule-binding repeats exaggerated. Features within the tau proteins and antibody epitopes: (N) inserts of acidic N-terminal domain (yellow); (P) proline-rich domain (green); (R) microtubule binding domains (MTBDs, blue) with four imperfect repeat regions separated by flanking sequences are not to scale, with microtubule-binding repeats exaggerated; C-terminal tail (grey); Eu, europium label; K18-recombinant tau fragment; RT QuIC- Real-Time Quaking-Induced Conversion. **(d)** Primary mouse cortical neurons inoculated at 7 DIV for 14 days and RA-differentiated SH-SY5Y cells inoculated for 3 days with 6 different AD brain-derived tau were applied as cell-based systems to evaluate tau seeding and transmission rates by conformation-dependent immunoassay (CDI), western blot (WB), and confocal laser scanning microscopy (CLSM); PID – post-inoculation days.

**Table 1. Demographics and descriptive parameters of AD cases.**

| Parameter | | Unit | n | Min | Max | Mean | ± | S.E.M. |
|---|---|---|---|---|---|---|---|---|
| Sex | | F/M | 17F/5M | | | | | |
| Age | | years | 22 | 43 | 91 | 70.4 | ± | 2.5 |
| PMI | | hrs | 20 | 4 | 72 | 33.8 | ± | 5.2 |
| Dis. Duration | Charts | month | 21 | 1 | 168 | 49.8 | ± | 10.7 |
| Neuropath Severity | A | Class | 22 | 2 | 3 | 2.7 | ± | 0.1 |
| | B | Class | 22 | 2 | 3 | 2.9 | ± | 0.1 |
| | C | Class | 22 | 1 | 3 | 2.3 | ± | 0.4 |
| APOE allele frequency | e2 | n(%) | 0(0) | | | | | |
| | e3 | n(%) | 27(61.4) | | | | | |
| | e4 | n(%) | 17(38.6) | | | | | |
| Amyloid beta | 1-42 | ng/ml | 16 | 104.3 | 1761.7 | 609.4 | ± | 114.7 |
| | | D/N ratio | 16 | 3.1 | 34.4 | 16.4 | ± | 2.5 |
| Tau protein | Insoluble | ug/ml | 22 | 86.3 | 2735.1 | 1187.9 | ± | 189.0 |
| | | D/N ratio | 14 | 23.5 | 406.9 | 108.0 | ± | 30.6 |

(D/N ratio) of Sarkosyl-insoluble tau protein monitored with direct conformation-dependent immunoassay (CDI, Table 1) showed major inter-individual differences that we observed previously [3].

### Validation of photochemical hydroxyl radical (•OH) footprinting of AD brain-derived tau conformers

Because our recent data showed major differences in seeding potency *in vitro* and propagation of distinct conformers of insoluble four-repeat AD tau aggregates in cells [3,27], we hypothesized that the structural organization of the microtubule-binding domain (MTBD) of misfolded tau could generate different conformers of misfolded tau (strains), which in turn could affect the rates of cognitive decline. Hydroxyl radical (•OH) protein footprinting is a structural covalent labeling technique that uses •OH generated *in situ* to probe the structure of monomeric, oligomeric, or fibrillar proteins [28–31]. In our recent investigation of human prions, we expanded the footprinting methodology to a new application that exploited the loss of antibody affinity due to the progressive hydroxylation of amino acid side chains within each particular antibody epitope [46]. This new ultrasensitive tool demonstrated major structural differences and, for the first time, revealed domains critical for replication and inactivation in the three most frequent human brain-derived prions [46].

Originally reported footprinting uses high-intensity X-rays in a synchrotron facility [46], but alternative methods generating •OH include Fenton's reaction and UV photolysis of hydrogen peroxide at 254nm [47]. For monitoring the hydroxylation impact in tau protein, we selected five monoclonal antibodies (mAb) with epitopes within all four repeat domains of the tau core and C-terminal tail (Fig 1C). Labeled with Eu, the mAbs demonstrated similar pg/ml sensitivity and a dynamic range of four orders of magnitude in detecting the monomers of the unfolded recombinant tau441 protein (rectau441) in CDI (S1A Fig). The hydroxyl radicals generated by UV irradiation of $H_2O_2$ covalently modified all accessible amino acid

side chains, and all five antibodies lost 99.5-99.9% of their affinity to rectau441 in the first 2 min (green diamonds in S1B, S1C, S1D, S1E, and S1F Fig). We used as a hydroxylation control recombinant tau441 instead of the alternative non-neurological, age-matched human brain because the reaction is quenched by a large excess of other proteins and Fe++ in human brain homogenates. Both normal monomeric brain tau and rectau441 (Fig 1) have open (unstructured) conformations [48]; if misfolded tau is present in age-matched non-neurological brain homogenates, its trace levels are not detectable with our Europium-labeled antibodies [3]. Under the conditions optimized for the differentiation of conformers of misfolded tau, the hydroxylation rate of tau441 monomers was 30–200-fold faster than that of insoluble aggregates of AD brain-derived tau, and all epitopes were fully hydroxylated in the first two minutes (green line in S1B, S1C, S1D, S1E, and S1F Fig). Rapid hydroxylation of tau441 monomers requires hydroxyl radical exposure on the millisecond time scale and a fully automated synchrotron hydroxylation station (S3B Fig). Mass spectrometry (MS) monitoring of footprinting performed at high concentrations of hydroxyl radicals generated in synchrotrons does not indicate significant protein cross-linking [28,30,31,46,49–54]. Additionally, our experiments with monomeric recombinant tau441 in the open conformation indicated progressive loss of antibody affinity on WBs with no concurrent crosslinking of tau monomers in silver staining (S3B Fig). We did not observe detectable concentrations of cross-linked peptides by MS in rectau441 monomers or AD brain-derived aggregates (S1G Fig) and coefficient of variation (($100 \times$ SEM)/Mean) of duplicate footprinting assay is ≤ 7%.

Cumulatively, the data indicated that in the recombinant tau441 monomers, the epitopes of all antibodies decayed at the same rate. Based on our understanding of •OH chemistry, the data are consistent with all core residues being fully exposed to the solvent in an open conformation of the rectau441 monomer [30,46]. In striking contrast, hydroxylation kinetics showed major and differential protection of distinct epitopes in two typical AD cases with different conformational stability assay (CSA) profiles. The half-life ($t_{1/2}$) determined by fitting the decay data with the Hill model shows a 30–200-fold extended half-life and distinctly different protection profiles of tau purified from AD brains with different CSA profiles (S1 Fig). The most striking difference in the hydroxylation rate was observed in the fourth domain (R4), with a lesser effect in R3 MTBD (S1D and S1E Fig). To collect a representative spectrum of detergent-insoluble, fibrillar and nonfibrillar, tau aggregates accumulating in the AD brain, we adopted the protocol used successfully in purification of human prions [35,46]. This protocol minimizes the nonspecific effects of human brain lysate components without proteases and allows us to investigate the same misfolded tau conformers in four independent systems simultaneously: (i) hydroxylation footprinting, (ii) RT QuIC for seeding potency [3], (iii) induction of tau pathology in mouse neurons [27], and (iv) induction of tau pathology and calcium flux in human neuronally differentiated SH-SY5Y cells [55]. Apart from the hydroxylation protocol, our tau-seeding protocols have been previously validated [3,27,55]. Atomic force microscopy (AFM) showed morphological heterogeneity of preparations with variable contents of paired helical filaments (PHs) [56], and smaller aggregates (S2A,S2B, S2C, and S2D Fig), and the overlapping conformational stability assay (CSA) [3,57] profiles of misfolded tau demonstrated that the final sample used in the hydroxylation profile protocol is representative of the tau conformers present in the brain homogenate (S2E Fig). SDS PAGE showed expected heterogeneity [58] with a variable fraction of SDS-resistant aggregates and oligomers, and an excess of four repeats (4R) over three repeat (3R) tau isoforms (S2F Fig) previously reported in the hippocampal cortex [3]. Cumulatively, initial validation experiments indicated variable •OH exposure of different epitopes in the MTBDs of different conformers of AD tau [46].

To independently investigate the antibody epitope decay effects induced by •OH, we analyzed the samples shown in S1G Fig after 8 min. photochemical (UV 254nm/ $H_2O_2$) hydroxylation using mass spectrometry [28,30,46,52]. Considering the impact of post-translational modifications [19,58], we optimized the digestion protocol using Asp-N (aspartate-N-Endo proteinase that hydrolyzes peptide bonds on the N-terminal side of Asp, and to a lesser extent Glu) and recombinant trypsin. Our data indicates that this sequential dual protease protocol generates reproducible peptide and posttranslational modification (PTM) coverage of brain-derived tau proteins downstream of residue ~240, including the C-terminal tail. The major differences in the solvent-exposed domains between rectau441 and representative AD cases (S1G Fig), and

differences in hydroxylation of specific residues corresponded to those found within the epitopes monitored with antibodies were observed (S1B, S1C, S1D, S1E, and S1F Fig). Although the progressive loss of antibody affinity to •OH-modified epitopes was detectable with WBs, SDS-insoluble high-mass oligomers of tau and a limited ~20-fold dynamic range precluded quantitative evaluation (S3 Fig).

## Conformational diversity of brain-derived tau epitopes in AD monitored by photochemical hydroxyl radical footprinting

We assessed the progressive hydroxylation of each of the five epitopes with insoluble tau enriched from the frontal cortex of 22 AD cases (S4 Fig). The data obtained with the five monoclonal antibodies demonstrated protection against hydroxylation in all four MTBDs and C-terminal tails, and major inter-individual differences in hydroxylation kinetics in all epitopes. Comparison of the hydroxylation rate in different epitopes indicated that the C-terminal tail was the least protected in all AD cases, followed by more protection in the R1, R2, and R3 domains, and the most variably protected fourth MTBD (R4) epitope (S4F Fig).

To compare the half-life hydroxylation profile for each epitope, we fitted the hydroxylation kinetics data in individual cases of AD with the Hill model, and the resulting epitope half-life profiles separated by the AD duration were plotted in polar plots (Fig 2). Within the spectrum of Alzheimer's disease phenotypes, the rapidly progressive subtype is characterized by rapid cognitive decline [4,7,11], a higher frequency of missense variants in known AD risk genes [1], a low frequency of the e4 allele of the APOE gene [4], and an accumulation of amyloid beta and misfolded tau proteins with distinct conformational characteristics [3,27,50] and interactomes [7,13,14]. Using the consensus criteria of more than six Mini-Mental State Examination (MMSE) points per year or death within 3 years of initial neurological diagnosis of dementia [1,4,7,11,59], the data indicated that the R4 domain was more protected from hydroxylation in a number of cases with longer disease durations (Fig 2A and 2B). However, this trend was not statistically significant due to the large data variance in each group. Notably, the variance in half-life observed in the R4 domain exceeded the C-terminal tail variance by ~9-fold. We interpreted the data as evidence of a distinct pattern of MTBD epitope exposure in individual AD tau conformers, implicating major inter-individual differences in the structural organization of the MTBD in AD brain-derived tau. Within the distinct patterns of protection in each of the conformer domains, photochemical footprinting provided direct evidence for the major variability in the fourth repeat (R4) of MTBD.

## Differential seeding potency of tau accumulation in the AD cortex

Evidence accumulated with prions and more recently with a growing list of other proteins indicate that the misfolded conformers can be amplified *in vitro* in a seeded reaction, a prerequisite step for misfolding propagation in the brain [24,50,60–62]. To establish the seeding potency and kinetics of misfolded tau propagation *in vitro*, we adapted this conformational templating mechanism for the amplification of AD brain-derived tau seeds using recombinant K18 (4R) tau substrates [48,63] (Fig 3A). Our initial titration experiments of both substrates and low molecular weight (LMW) heparin cofactor using a 60 hr experimental window were designed to obviate the spontaneous formation of Thioflavin T-positive aggregates and to obtain maximized responses from homologous brain-derived tau seeds. At the same recombinant K18 substrate concentrations, tau seeds preferentially amplified with homologous substrates and demonstrated end-point sensitivity up to 10-8 dilution of the brain tissue, corresponding to an average of 60 fg of misfolded tau per well, and lower sensitivity with a stochastic response with non-homologous substrates [3]. With the K18 (4R) substrate, the specific seeding activity expressed as lag phase per ng of frontal cortex-derived tau demonstrated differences between rapidly and slowly progressive AD cases (Fig 3B) we observed previously with hippocampal cortex tau [3]. We interpret these data as evidence of the broad seeding potency of distinct conformers of tau in individual AD cases, with more potent seeds accumulating in AD cases with faster clinical progression [3].

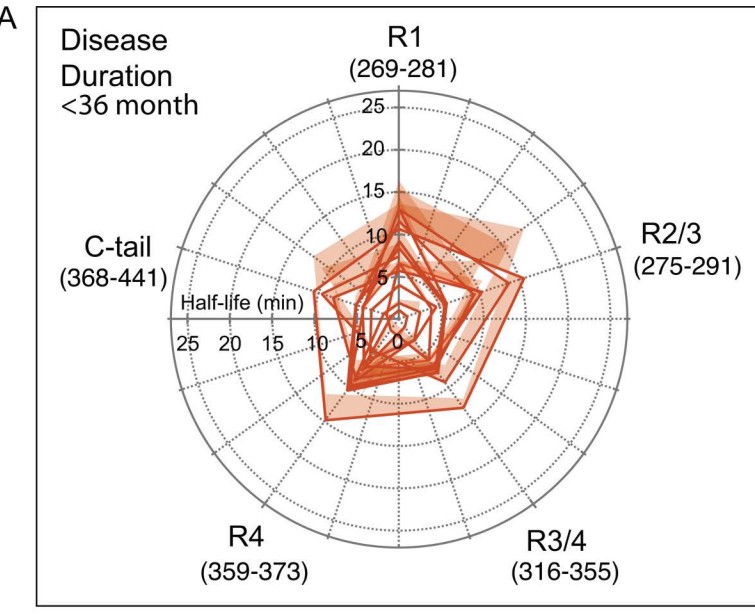

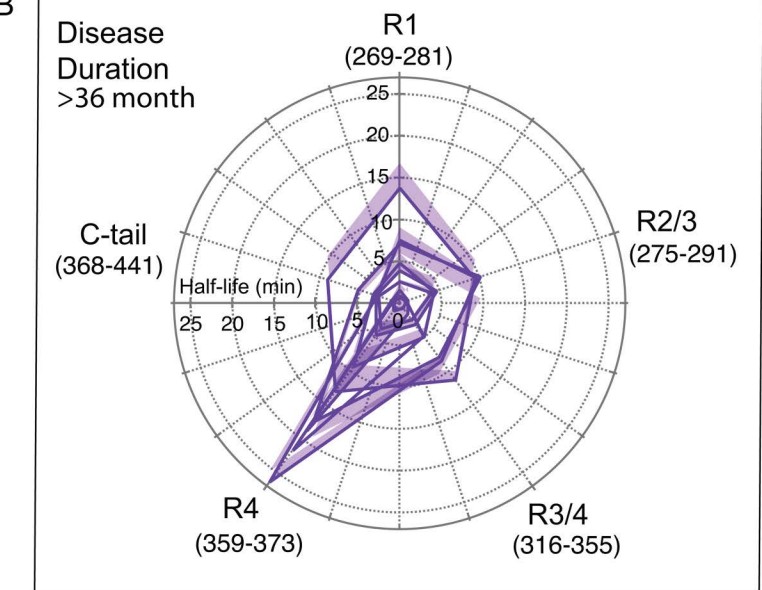

**Fig 2. Half-lives of different MTBDs and C-tail epitopes in tau isolated from frontal cortex of AD cases with different disease duration. (A,B)** Each curve and shade are an average±**S**.E.M. obtained from duplicate photochemical footprinting experiments for Sarkosyl-insoluble tau isolated from one AD case (n = 22). The domains are labeled as outlined in Fig 1 with the epitope amino acid sequence in the parenthesis.

Next, we tested the effect of each epitope of the Sarkosyl-insoluble MTBD of tau on the lag phase. To quantify this impact, we used linear regression models of the dependency of the specific seeding activity of tau on the half-life of each MTBD epitope. Fig 3C shows that the tau seeding activity correlated significantly (P = 0.0003) with the half-life of the R4 MTBD domain due to differences in the structural organization of protease-resistant Sarkosyl-insoluble tau conformers (Fig 3D). The regression analyses of in vitro RT QuIC seeding and hydroxylation data (Table 2) may suggest also significant contribution of R3 domain because the epitope of mAb 77G7 covers both R4 (19 amino acids) and R3 (16 amino

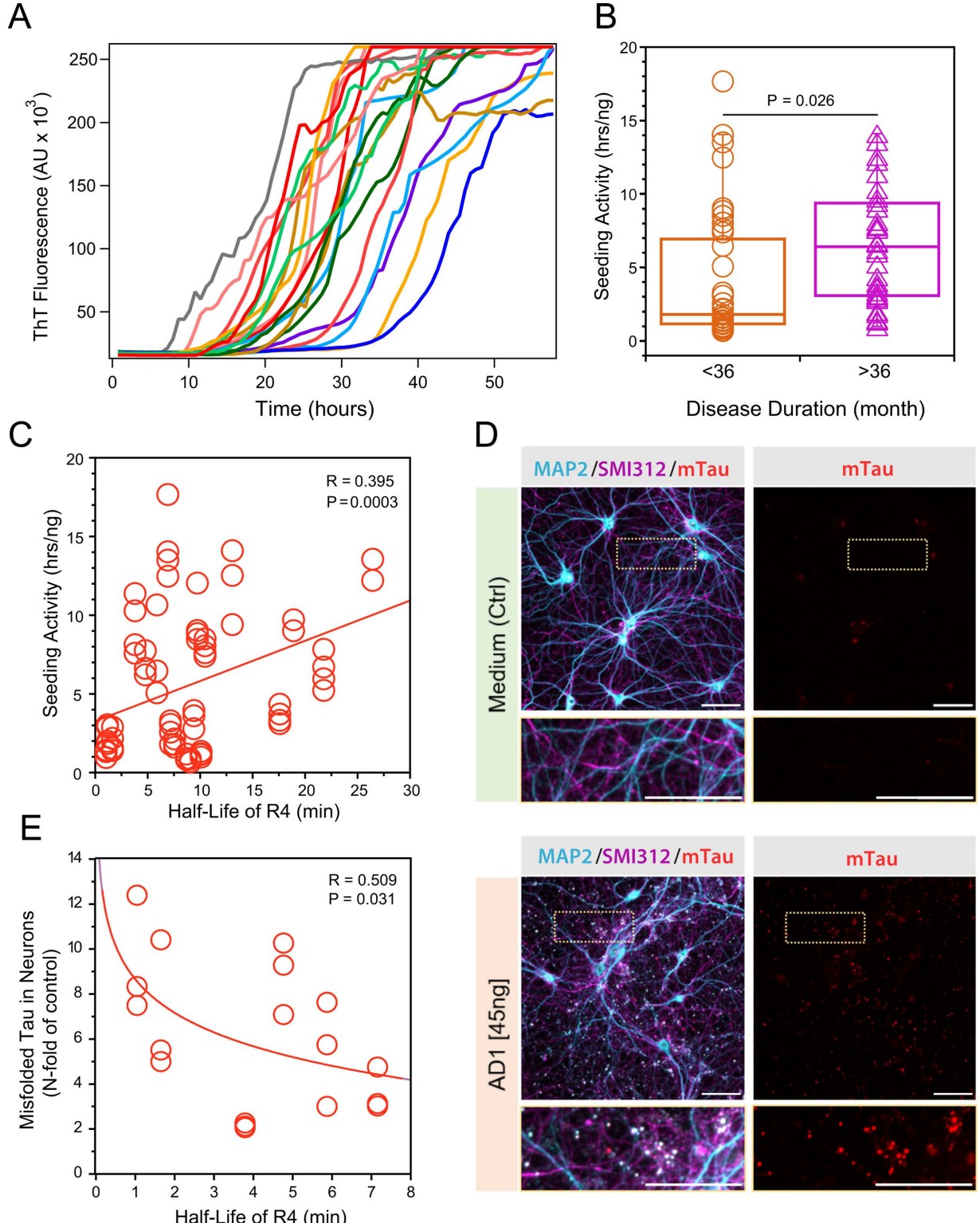

**Fig 3. Variable seeding potency and propagation rates of different conformers of tau. (A)** K18 substrates seeded with AD (n = 20) brain samples and monitored in real time by thioflavin T fluorescence; the samples were diluted 10⁴-fold, and the curves are average of Thioflavin T fluorescence intensity in four wells at a given time point and unseeded K18 constructs were applied as negative controls of spontaneous aggregation. **(B)** Seeding activity

of brain-derived AD tau (n = 20) expressed as lag phase (hrs) per ng; the lag phase was determined in four independent RT-QuIC seeding experiments. Each box encloses 50% of the data with median value displayed as a line, whiskers mark the minimum and maximum, and individual point indicate an outlier value outside the UQ + 1.5*IQR interval, where UQ is upper quartal, and IQR is inter-quartal range. Statistical significance was determined with ANOVA. **(C)** Linear regression model of seeding activity dependency on R4 domain half-life in individual AD brain samples (n = 20). **(D)** Representative confocal microscopy images of accumulating aggregates of mouse tau in mature neurons exposed to a single dose of AD brain-derived human tau (45ng/well) fourteen days earlier. The neurons were stained with antibodies specific to mouse tau (red), somatodendritic marker MAP2 (cyan), and axonal neurofilaments (SM1312, purple). **(E)** Regression analysis of concentration of Sarkosyl-insoluble tau accumulating in neurons and R4 domain half-life of AD brain-derived tau (n = 6). The neuronal accumulation of Sarkosyl-insoluble tau was measured by CDI in three different experiments.

**Table 2. Regression analysis of the impact of MTBD half-life on lag phase of seeding in RT QuIC.**

| Eu-labeled mAb | MTBD | Correlation Coefficient | Significance |
|---|---|---|---|
| | Epitope (AA residues) | R | P |
| 16040D | R1 (269–281) | 0.2766 | 0.0130 |
| RD4 | R2/3 (275–291) | 0.2660 | 0.0171 |
| 77G7 | R3/4 (316–355) | 0.3660 | 0.0008 |
| 16097B | R4 (359–373) | 0.3947 | 0.0003 |
| 16097F | C-tail (368–441) | 0.2292 | 0.0408 |

acids) domains, but to discern the contribution of R3 domain will require high resolution synchrotron hydroxylation footprinting monitored with mass spectrometry. Nevertheless, the seeding data in mice neuronal cultures (Fig 3E) and neuronally differentiated human cells (Figs 4 and 5D) consistently implicate R4, not R3 domain, as the significant driver of seeding in vivo. The impact of the remaining domains was less significant, and whether these effects reflect the size of the assembly of tau conformers or specific structural arrangements will require sucrose gradient or chromatographic profiling [3] (Table 2). Cumulatively, the data indicates that the reliable predictor of seeding activity is the exposure and structural organization of the R4 domain, with contribution of the R3 domain, and lesser effect of the remaining domains and the C-terminal tail.

### Seeding activity of brain-derived tau in AD measured in primary mouse neurons

To investigate the tau propagation tendency of distinct tau conformers, we inoculated wild-type primary mouse neurons with structurally characterized Sarkosyl-insoluble tau isolates from the frontal cortex of six AD cases and monitored the accumulation of de novo-induced tau aggregates using confocal microscopy (Fig 3D) and CDI (Fig 3E) [27]. At the same concentrations of AD brain-derived tau, different AD tau isolates induced the accumulation of misfolded mouse tau aggregates at different levels, and previous experiments demonstrated distinct conformational characteristics corresponding to the original AD brain tau [27]. By applying CDI (Fig 3E), the concentration of misfolded insoluble tau accumulated in neurons inversely correlated with the half-life of the four-repeat (4R) tau domain in the six different AD brain-derived tau's (P = 0.031). Taken together, the newly formed tau aggregates in mature mouse neurons expressing physiological levels of mouse four-repeat tau showed a dependency of propagation on the structural characteristics of MTBDs, with the least protected R4 domain driving the speed.

### Misfolding, transmission rate, and toxicity of tau conformers in RA-differentiated SH-SY5Y

To investigate the replication and propagation of misfolded tau conformers in neuronal cultures expressing physiological levels of human tau protein, we inoculated RA-differentiated SH-SY5Y cells with six distinct conformers of AD brain-derived tau. The protocol including gradual serum starvation and addition of all-trans-Retionic acid (RA) converts

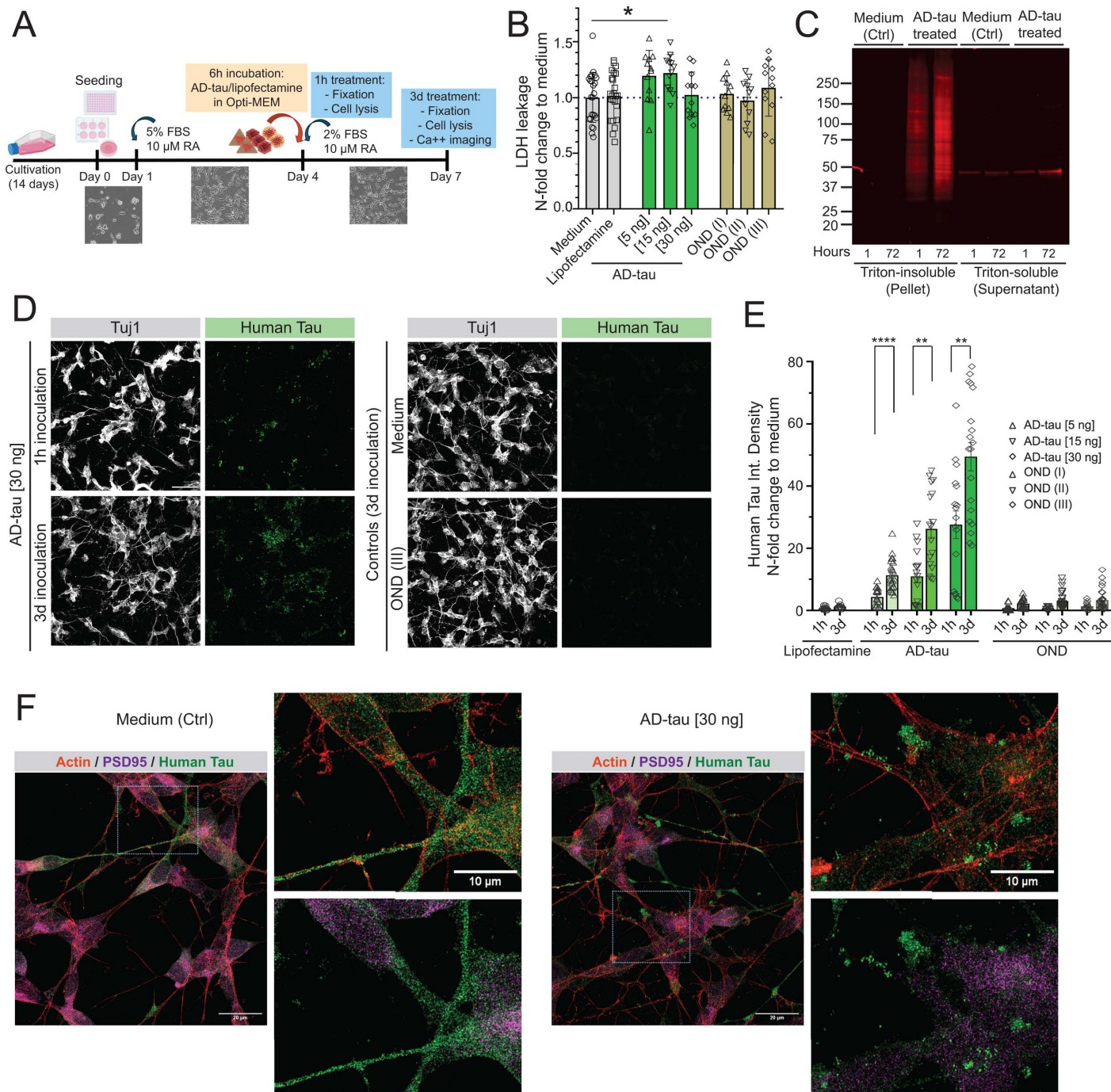

**Fig 4. Propagation of tau protein misfolding in RA-differentiated SH-SY5Y cells expressing mature human tau protein. (A)** The timeline of RA-induced differentiation of SH-SY5Y neuroblastoma cells and their inoculation with AD brain-derived tau. **(B)** LDH assay of RA-differentiated SH-SY5Y cells inoculated with AD brain-derived tau in three concentrations (5, 15, and 30 ng/ well in 96 well-plate) and with controls (other neurological disorders, OND) in corresponding volume as tau per well. The LDH levels in media were measured from at least five wells in two independent experiments. Each box encloses the data with mean value ± SEM and individual points indicate N-fold change to non-treated wells (medium only, Ctrl). Statistical significance was determined with one-way ANOVA. **(C)** Western blot with triton-soluble and insoluble fractions of cell lysates of differentiated SH-SY5Y cells inoculated with AD brain-derived tau or non-treated (medium, Ctrl) for 1 and 72 hours. **(D)** Representative confocal images of max intensity projections of 15 z-stacks (0.40 μm each) of RA-differentiated SH-SY5Y inoculated with AD brain-derived tau (30ng/well) for 1 hour and 3 days, OND (* corresponding volume added) for 3 days and cells in medium only. Neuronal marker Tuj1 (grey), human tau (clone RTM49, aa2-44, green) in 1% Triton-fixed cells,

scale bar is 50 µm. **(E)** Integrated Density of tau fluorescence expressed as x-fold to medium levels were measured in wells inoculated with AD brain-derived tau for 1 h and 3 d in three concentrations (5, 15, and 30 ng) and with OND control added in corresponding volumes. Cultures inoculated with no inoculates (lipofectamine) were included as controls. Each box encloses the data with mean value ± SEM and individual points indicate N-fold change to non-treated wells (medium only, Ctrl). Statistical significance was determined with student t-test (P-values: ** P ≤ 0.01, **** P ≤ 0.0001). At least three areas from two wells in three independent experiments were captured. **(F)** Representative stimulated emission depletion (STED) microscopy images of max projections of 12 z-stacks (0.35 µm each) after deconvolution and brightness corrections show actin (red), PSD95 (magenta), and human tau (green) in non-treated cultures (medium, control) and inoculated with AD brain-derived tau. Scale bar is 20 µm, cropped images 10 µm.

SH-SY5Y obtained from the American Type Culture Collection (ATCC) solely composed of neuroblastic N-type cells into neuron-like cells with distinctly polarized axon-dendritic morphology observed with immunostaining against tau, MAP2 and Tuj1 (class III β-tubulin) (S5A Fig). Differentiated SH-SY5Y form bona fide synaptic connections [55,64], with synaptophysin and PSD95 proteins sorted into distinct cellular compartments (S5B Fig), and show calcium influx when challenging with a calcium ionophore ionomycin (S5C and S5D Fig). Exposure of RA-differentiated SH-SY5Y cells to misfolded AD brain-derived tau proteins at day 4 (Fig 4A) induced progressive accumulation of triton-insoluble aggregates of tau proteins in a concentration-dependent manner (Fig 4C, 4D, and 4E) with no apparent direct cytotoxicity as indicated by no significant change in levels of lactate dehydrogenase (LDH) leakage into media (Fig 4B). The exposure of differentiated neuron-like SH-SY5Y cells to AD brain-derived tau triggered a major change in tau staining pattern with aggregates of misfolded tau replacing the original uniformly diffuse localization of tau monomers (Fig 4F). The missorting accompanying protein misfolding was specific to tau protein as PSD95 uniformly dispersed pattern remained unchanged (Fig 4F). The differential cell impacts of human AD brain-derived insoluble tau at the same concentration determined after complete denaturation with 4M Gdn HCl by conformation-dependent immunoassay (CDI) indicate distinct structural organization of each isolate and interactome epitopes. Using the same sample in two different cell systems expressing full length mice and homologous human tau implicate structural exposure of the R4 domain in seeding and propagation and thus confirm the in vitro seeding data obtained with K18 tau fragments in RT QuIC.

The immunostaining of RA-differentiated SH-SY5Y cultures with tau R1 MTBD-specific (clone 16040D, 269–281 aa) and hyperphosphorylated tau-specific AT8 antibody revealed variability among individual AD brain-derived tau seeds to trigger tau misfolding and pathological phosphorylation with a delay in the deposition of tau aggregates phosphorylated at Ser202 and Thr205 (Figs 5A, 5B, 5E, 5F, and S6C). The accumulation rates of tau aggregates in SH-SY5Y and mouse neuron cultures (S5C Fig) were comparable (P = 0.001) and demonstrate that in both neuronal systems are driven primarily by the characteristics of AD brain-derived tau conformers. As with mouse neurons, the structural exposure of the R4 MTBD was the most significant driver of tau propagation in SH-SY5Y cells (P = 0.0008) (Fig 5D). In contrast to no detectable general cytotoxicity (Fig 4B), KCl-induced depolarization (Fig 5G) demonstrates various degree of calcium influx dysregulation in SH-SY5Y inoculated 3 days with individual AD brain-derived tau seeds (S6A, S6B, S6D, and S6E Fig), likely linked to excitotoxicity [65,66], with the most significant impact of AD brain-derived conformers with structurally exposed R1 domain (P = 0.006) (Fig 5H).

## Discussion

Our recent findings implicated distinct and highly potent tau seed conformers in the rapid progression of AD [3,27]. Our working hypothesis was that the specific structural organization of microtubule-binding domains (MTBDs) in pathogenic tau conformers is a critical driver of their replication by controlling the affinity to monomers of normal tau, thus leading to different rates of replication and propagation in neurons. To test this hypothesis and the impact of different structural organization of distinct conformers of AD brain-derived tau, we introduced a new photochemical hydroxylation footprinting protocol monitored with a panel of Eu-labeled antibodies. In tandem with seeding and propagation rates determined *in vitro* and in neuronal cultures, the data demonstrated (i) major inter-individual structural diversity of MTBDs in tau

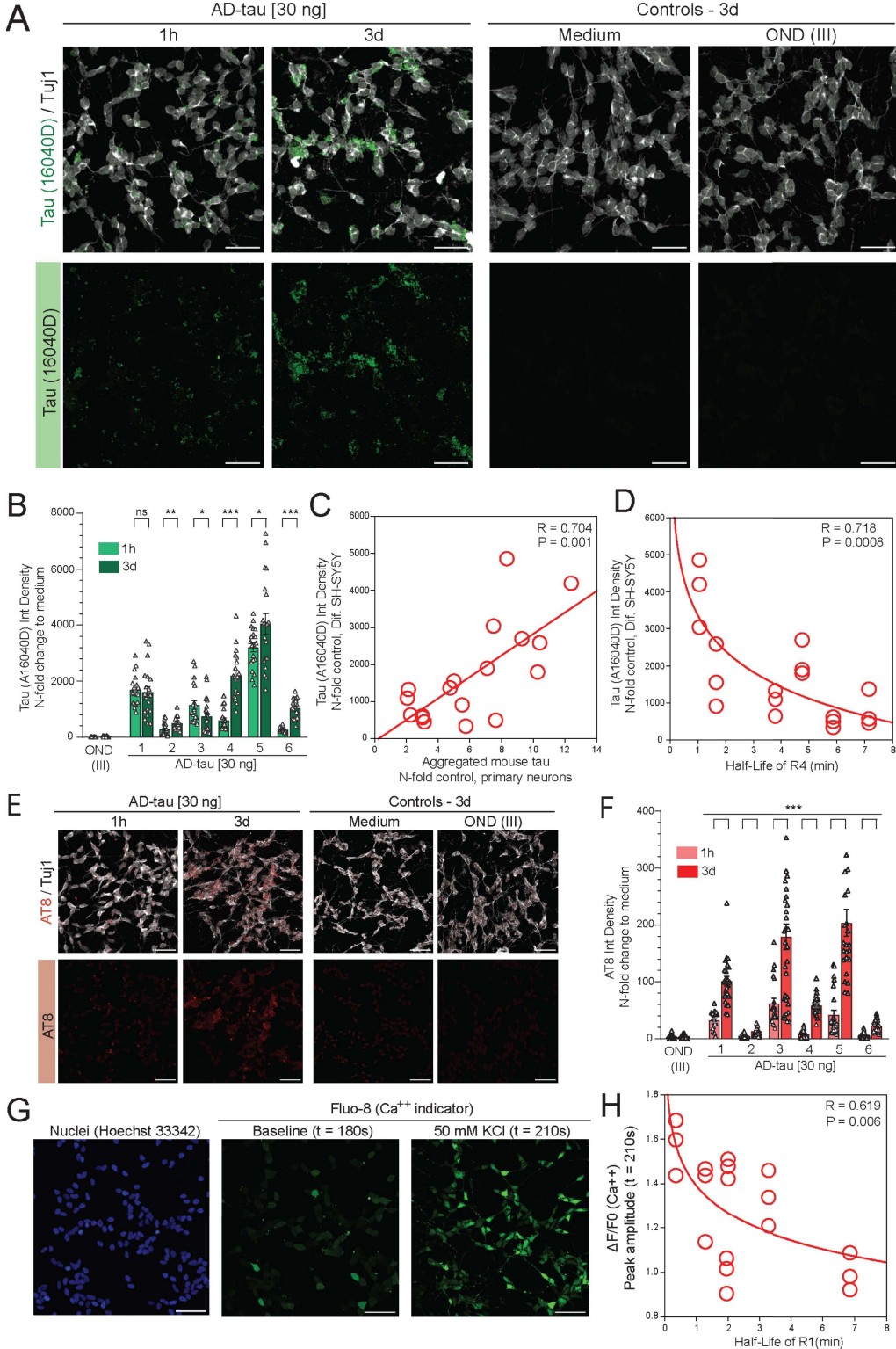

**Fig 5. Transmission rate of human tau misfolding and calcium homeostasis in differentiated SH-SY5Y cells inoculated with AD brain-derived tau. (A)** Representative confocal images of max intensity projections of 15 z-stacks (0.40 μm each) of RA-differentiated SH-SY5Y inoculated with AD brain-derived tau (30ng/well) for 1 hour and 3 days, OND (* corresponding volume added) for 3 days and cells in medium only. Neuronal marker Tuj1

(grey), tau (clone 16040D, aa269-281, green) in 1% Triton-fixed cells, scale bar is 50 μm. **(B)** Integrated Density of tau fluorescence expressed as x-fold to medium levels. Each box encloses the data with mean value ± SEM and individual points indicate N-fold change to non-treated wells (medium only, Ctrl). Statistical significance was determined with student t-test (P-values: * $P \leq 0.05$, ** $P \leq 0.01$, *** $P \leq 0.001$). At least three areas from two wells in three independent experiments were captured. **(C)** Linear regression analysis of levels of IntDensity fluorescence of tau in differentiated SH-SY5Y and concentration of Sarkosyl-insoluble tau accumulating in mouse primary neurons both inoculated with AD brain-derived tau (n = 6). **(D)** Non-linear regression analysis of levels of IntDensity fluorescence of tau in differentiated SH-SY5Y inoculated with AD brain-derived tau for 3 days and R4 domain half-life of AD brain-derived tau (n = 6). **(E)** Representative confocal images of max intensity projections of 15 z-stacks (0.40 μm each) of RA-differentiated SH-SY5Y inoculated with AD brain-derived tau (30ng/well) for 1 hour and 3 days, OND (* corresponding volume added) for 3 days and cells in medium only. Neuronal marker Tuj1 (grey), hyperphosphorylated tau (clone AT8, red) in 1% Triton-fixed cells, scale bar is 50 μm. **(F)** Integrated Density of AT8-positive tau fluorescence expressed as x-fold to medium levels. Each box encloses the data with mean value ± SEM and individual points indicate N-fold change to non-treated wells (medium only, Ctrl). Statistical significance was determined with student t-test (** $P \leq 0.001$). At least three areas from two wells in three independent experiments were captured. **(G)** Representative confocal images of nuclei staining (Hoechst 33342, blue), baseline Fluo-8 immunofluorescence at 180 s time point and at the fluorescence peak at 210 s, 30 s after KCl-induced calcium influx. Scale bar is 50 μm. **(H)** Non-linear regression analysis of Fluo-8 fluorescence intensity in ratio to baseline levels at max peak (medium controls, t = 210 s) in differentiated SH-SY5Y inoculated with AD brain-derived tau for 3 days and R1 domain half-life of AD brain-derived tau (n = 6).

conformers isolated from AD with different progression rates, (ii) striking variability in the fourth repeat (R4) tau domain, and (iii) a significant role of the structural organization and exposure of the R4 domain within the MTBDs in the replication and propagation of tau conformers (Fig 6).

The UV photolysis of $H_2O_2$ allows for rapid high-throughput covalent modifications of solvent-exposed amino acid side chains with hydroxyl radicals (•OH) on a benchtop. The progressive decay of the antibody epitope due to photochemical hydroxylation offers rapid monitoring of the structural organization in all MTBDs, C-terminal tail, and other important domains in misfolded tau aggregates with high-affinity Eu-labeled antibodies. The advantages of this approach include high throughput, which allows the simultaneous monitoring of multiple epitopes in several samples. Another major advantage of our method over hydrogen/deuterium exchange monitored by mass spectrometry is that (i) hydroxylation provides stable covalent modification (no back-exchange) compatible with time- and pH-independent standard downstream protocols, and (ii) allows plate-formatted monitoring of epitope hydroxylation with Eu-labeled antibodies in a high-throughput 96-well plate protocol. The identification of structural epitopes driving the important effects of oligomeric and fibrillar misfolded tau structures should facilitate the differential phenotype-based diagnostics of AD and the development of tau inhibitors. Moreover, analysis of the targeted epitopes using mass spectrometry will increase the resolution to a single

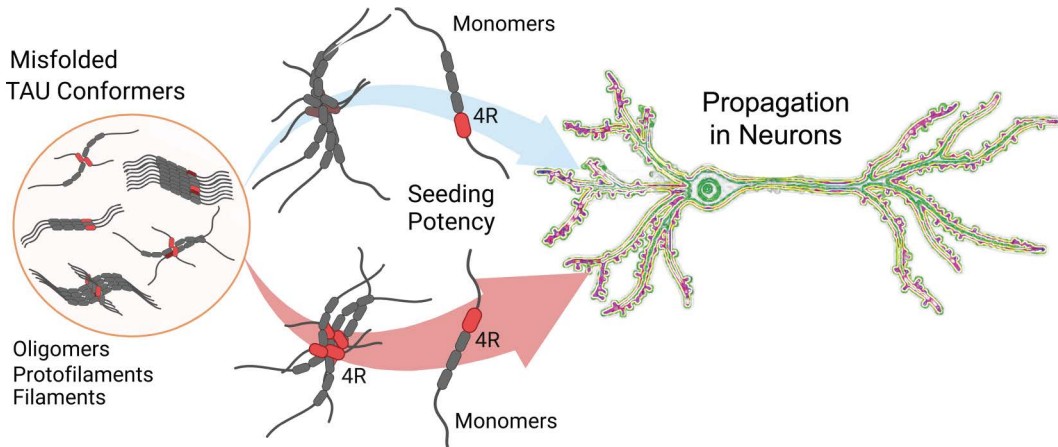

**Fig 6. Graphical abstract.** The role of variable structural exposure of fourth repeat tau domain (R4, red) within the MTBDs in the propagation of tau conformers in neuronal cultures.

amino acid [46], thus facilitating the design of high-affinity monoclonal antibodies specific for distinct conformers or small-molecule inhibitors.

Because all six isoforms of tau are equally expressed in an adult human brain, the assumption was that all six will form neurofibrillary tangles (NFTs) and that isoforms are probably randomly integrated into paired helical filaments (PHFs) and straight filaments (SFs) in AD [21,67,68]. However, our early separation by high-speed centrifugation in sucrose gradient showed that misfolded tau aggregates are, regardless of size, composed uniformly of ~80% of 4R tau and ~20% of 3R tau, even though the normal soluble tau monomers are mixtures of approximately equal parts of 3R and 4R isoforms [3]. The 4-fold excess of 4R tau in misfolded tau aggregates suggests that 4R tau monomers have a higher propensity for conformational transition to beta-sheet structures in AD. Hydroxylation footprinting provided direct evidence for the critical role of the distinct structural organization and exposure of the fourth repeat (R4) in AD brain-derived tau to solvent, and presumably to tau monomers, in driving seeding *in vitro* and propagation in neuronal cultures.

To validate the data in the cell system expressing physiological levels of mature human tau forms, we exposed the differentiated SH-SY5Y neuronal cell model to AD brain-derived tau. The variable exposure of R4 domain drove the conformational transition of monomeric human tau with patient-to-patient variability and without significant global cytotoxic effects. However, the calcium influx imaging showed already major changes suggesting functional effect that may lead to synaptic toxicity [65] and that we observed previously in differentiated mouse neurons [27]. Pathological tau disrupts calcium homeostasis in neurons [69–73], impairing neuronal functions and leading to excitotoxicity through excessive calcium influxes [66]. In this study, differentiated SH-SY5Y cells inoculated with six different AD brain-derived tau exhibited a high degree of inoculum-to-inoculum variability in calcium influx. Notably, elevated intracellular calcium levels following depolarization correlated with the exposure of the R1 domain in tau strains. Cultured neurons exposed to pathological tau have been shown to increase intracellular calcium levels [65,69,71,74,75] and our data point to the specific effect of R1 MTBD domain in misfolded tau conformers. Future studies will focus on identifying neuronal interactors with R1 domain and the mechanism driving differential effects.

Numerous posttranslational modifications of tau add another layer to the conformational complexity of brain-derived tau, and using mass spectrometry, several papers recently reported the differences in profiles of posttranslational modifications (PTMs) of insoluble tau that correlated with the seeding potency of insoluble tau in different clinical phenotypes of late-onset AD [58,76]. Whether the differences in conformation drive distinct PTMs, or vice versa, is an important question not only in AD but also in other tauopathies, including progressive supranuclear palsy (PSP), frontotemporal dementia (FTD), and Pick's disease. Integrating photochemical hydroxylation footprinting with mass spectrometry of tau will address this question by allowing simultaneous investigation of both aspects. These approaches are applicable to both filamentous and oligomeric structures and should also answer the question of whether the differences in hydroxylation footprinting are solely a result of conformation and particle size or whether they are modified by Sarkosyl-resistant ligands or attached lipids [35,46,50,60]. New evidence that cryo-EM structures of prions [77], and tau filaments [19,21,68], isolated from animal and human brain are impacted by PTMs and retain auxiliary, non-proteinaceous densities suggests that such associations are more broadly maintained across different filament extraction protocols from the human brain and play a role in their formation and interactions. To assign the structural interplay between PTMs and auxiliary noncovalent interactors to different domains [76] in fibrillar and nonfibrillar assemblies will require separation of different aggregates by sucrose gradients or HPLC integrated with synchrotron hydroxylation and mass spectrometry to monitor high resolution structure and PTMs in different tau domains, and then correlating the data with cryo-EM.

## Materials and methods

### Ethics statement

All procedures were performed under protocols approved by the Institutional Review Board of the Case Western Reserve University and University Hospitals Case Medical Center in Cleveland, OH, USA. In all cases, written informed consent

was obtained from the patient or legal guardian, and the materials used had appropriate ethical approval for use in this project. All patient data and samples were coded and handled according to the NIH guidelines to protect patient identities.

## Study design

This study aimed to investigate the structural organization and replication potency of misfolded aggregates of tau protein in the frontal cortex of patients with different progression rates of Alzheimer's disease (AD). The concentration and conformational characteristics of tau protein isoforms were determined in three independent experiments using biophysical methods based on our previous work [3,4,24,45,60]. The sample size for biophysical and time points for cell experiments were chosen based on data reported earlier [3,24] to obtain a significant difference in the expected variability. Deidentified and randomly assigned human postmortem brain tissue samples were collected from brain bank repositories located at Case Western Reserve University (Table 1). The identity of the samples was replaced with an internal code, and the investigators performing the experiments were blinded to sample identity during testing and analysis. The experimental replicates for each experiment are listed in the figure legends.

## Patients and clinical evaluations

To analyze the full spectrum of AD phenotypes, cases were randomly selected from two CWRU biobanks. The cases of rapidly progressive AD (rpAD) originated in a group of 439 patients with a definitive diagnosis of rapidly progressive sporadic AD who were referred to the National Prion Disease Pathology Surveillance Center (NPDPSC) between 2003 and 2017 with rapidly progressive dementia and a differential diagnosis of prion disease. In all cases, we were able to exclude familial or sporadic prion disease after sequencing the PRNP gene, conducting neuropathology and immunohistochemistry for the pathogenic prion protein (PrP$^{Sc}$), and molecular typing of PrP$^{Sc}$ using Western blots [46,62,78]. Case records accumulated by trained personnel using the standard NPDPSC protocol were retrospectively analyzed. These records included medical charts, semi-structured telephone interviews of prion surveillance center personnel with patients and caregivers at the time of referral, EEG, MRI, and laboratory results [62,78]. The criteria for inclusion in the rpAD cohort were: (1) initial referral to NPDPSC and classification as possible prion disease because of its clinical appearance in accordance with the consensus official criteria valid at the time of referral [4,10,78]; (2) decline in more than six Mini-Mental State Examination (MMSE) points per year or death within 3 years of initial neurological diagnosis of atypical dementia [7,10,59]; (3) absent autosomal dominant pattern of dementia; (4) absence of pathogenic mutations in the human prion protein (PrP) gene (*PRNP*); (5) neuropathology and immunohistochemistry of tau proteins and amyloid beta with unequivocal classification as sporadic AD; (6) absence of neuropathological comorbidity; and (7) distribution of means and proportions of demographic data within 95% confidence interval of the whole group, resulting in no difference in means and proportions between randomly selected and all AD cases in the NPDPSC database. Because there are no definite clinical criteria for rpAD [4,6,7,9,10] and to prevent contamination of this cohort with outliers, for further studies, we selected cases within the normal distribution interval of disease duration calculated as UQ + 1.5*IQR, where UQ is the upper quartile range and IQR is the inter-quartal range.

The classical slowly progressive AD cases (spAD) cases were defined as those diagnosed between 2001 and 2013 at the Brain Health and Memory Center of the Neurological Institute at University Hospitals Case Medical Center, and brains were collected in the repository of the Department of Pathology at Case Western Reserve University [3,4,32,79]. The criteria for inclusion in the spAD cohort were as follows: (1) unequivocal clinical diagnosis of AD [80]; (2) absence of an autosomal dominant pattern of dementia; (3) unequivocal classification of AD after detailed neuropathology and immunohistochemistry of tau proteins and amyloid beta using NIA-AA criteria [81,82]; (4) absence of concurrent clinical or neuropathologic comorbidity; and (5) distribution of means and proportions of demographic data within the 95% confidence interval of late-onset cases accumulated at the National Alzheimer's Coordinating Center (NACC) at the University of Washington between September 2005 and February 2013 [83]. In all cases, the clinical diagnoses of probable spAD

and rpAD were confirmed by diagnostic histopathology [80]. For comparison of disease durations, we used late-onset autopsy-proven AD cases submitted to the National Alzheimer's Coordinating Center (NACC) database at the University of Washington. The control age-matched non-neurological group consisted of age- and sex-matched patients whose primary causes of death were lymphoma, carcinomatosis, or autoimmune disorders, and whose neuropathology ruled out prion disease, AD, or other neurodegenerative disorders.

## Sequencing of PRNP, APOE, APP, PSEN1, and PSEN2 genes

DNA was extracted from frozen brain tissue in all cases, and genotypic analysis of *APOE* gene polymorphism and *PRNP* coding region was performed as previously described [84,85]. The coding regions of *APP*, *PSEN1,* and *PSEN2* were analyzed using a TruSeq Custom Amplicon kit generated by DesignStudio (www.Illumina.com), as reported previously [1,4]. Screening for *APOE* alleles and exons 4 and 5 in *PSEN1* was carried out using polymerase chain reaction followed by Sanger dideoxy sequencing, as described previously [1,4,86]. Frontotemporal lobar degeneration (FTLD)-MAPT P301L patients and Pick disease cases of both sexes have been described previously, and the clinical features of the patients were assessed according to contemporaneous criteria for diagnosis [24,87].

## Brain sampling and conformation-dependent immunoassays (CDI) for Aβ40 and Aβ42

Coronal sections of human brain tissues were obtained at autopsy and stored at -80ºC. Slices of the cortex weighing 200–350 mg were homogenized to a final 15% (w/v) concentration by three 75 s cycles with a Mini-Beadbeater 16 Cell Disrupter (Biospec) in Tris-buffered saline (TBS), pH 7.4, and stored at-80 °C for future analysis. The conformation-dependent immunoassay for Aβ40 and Aβ42 was based on the principles developed for the measurement and characterization of prions (Table 1) [40,88] as described previously [3,4].

## Sandwich-formatted CDI for misfolded tau conformers

Experiments were performed according to a previously described protocol [3,27]. Briefly, samples of frontal cortex homogenate were diluted to a final concentration of 10% (w/v) with PBS containing 2% Sarkosyl and centrifuged at 21,500 × g at 4°C for 30 min in an Allegra X-22R tabletop centrifuge (Beckman Coulter) to obtain Sarkosyl-soluble and insoluble tau fractions. The pellet containing Sarkosyl-insoluble tau protein was resuspended in PBS (pH 7.4), and CDI was performed as described previously for mammalian prions [40,45,85], amyloid beta [4], and recently for brain-derived tau [3,24,27] with minor modifications. First, we used white Lumitrac 600 high-binding plates (E&K Scientific) coated with mAb DA9 (epitope 102–139, gift of late Dr. Peter Davies) in 200 mM $NaH_2PO_4$ containing 0.03% (w/v) $NaN_3$, pH 7.5. Each sample was split into two aliquots: the first was denatured (D) with a final concentration of 4M Gdn HCl, and the second aliquot, native (N), was untreated. Aliquots of 20µl from each aliquot containing 0.007% (v/v) Patent Blue V (Sigma) were directly loaded into the wells of white strip plates prefilled with 200 µl of casein/ 0.05% Tween 20 in TBS (pH 7.4) (SurModics). Finally, the captured tau was detected by a Europium-conjugated [45] anti-tau mAb 77G7 (epitope 316–355 of 2N4R tau; BioLegend) and mAb RD3 (epitope 267–316, [89,90]) (Fig 1), and the time-resolved fluorescence (TRF) signals of europium were measured using a multimode microplate reader (PHERAstar Plus; BMG LabTech). Eu-N1 ITC (Perkin Elmer) labeling was performed as we described previously [45] with a final Eu/IgG molar ratio of 4.4 for 77G7 and 3.8 for RD3, respectively. Recombinant 2N4R (tau441) and 2N3R (tau410) splicing variants of human tau expressed in *E. Coli* (rPeptide, Watkinsville, GA, USA) without His-tag [91] were used as calibrants after complete denaturation in 4M Gdn HCl. The initial concentration of reduced recombinant human tau441 was calculated from the absorbance at 280 nm and molar extinction coefficient of 7450 M-1 cm-1. The purified recombinant proteins were dissolved in 4M Gdn HCl and 50% Stabilcoat (SurModics) and stored at -80°C. The concentration of total (3R + 4R) tau was calculated from the CDI signal of denatured samples detected with the Eu-77G7 mAb and a calibration curve created from serially diluted recombinant

tau441. The TRF signal of denatured and native sample aliquots was expressed as a ratio (D/N) and was a measure of exposed epitopes in the native state against the reference of the fully unfolded protein.

## Monitoring the dissociation and unfolding of misfolded tau conformers using conformational stability assay (CSA)

Sequential denaturation of human tau was performed as described previously for mammalian prions [40,45,60], following modifications described recently for tau [3,24,27]. The 5% brain homogenate in PBS containing 2% Sarkosyl was mixed with a protease inhibitors cocktail (0.5mM PMSF and aprotinin and leupeptin at 5 µg/ml, respectively) and precipitated with 0.64% of Sodium Phosphotungstate and 5mM $MgCl_2$ after incubation for one hour at 37°C in Eppendorf Thermomixer (Eppendorf) as described for prions and tau [3,40,45,92,93]. Pellets were collected at 21,500 × g and 4°C for 30 min. in an Allegra X-22R tabletop centrifuge (Beckman Coulter) and stored at -80°C.

Frozen aliquots of samples containing tau were thawed, sonicated for 3 × 5 s at 80% power with a Sonicator 4000 (Qsonica), and the concentration was adjusted to a constant ~3.5 µg/ml of tau. 15 µl aliquots in 15 tubes were treated with increasing concentrations of 8M Gdn HCl containing 0.007% (v/v) Patent Blue V (Sigma) in 0.25 M or 0.5 M increments. After 30 min incubation at room temperature, individual samples were rapidly diluted with casein/ 0.05% Tween 20 in TBS, pH 7.4 (SurModics) containing diminishing concentrations of 8M Gdn HCl, so that the final concentration in all samples was 0.411 M. Each individual aliquot was immediately loaded to dry white Lumitrac 600, High Binding Plates (E&K Scientific), coated with mAb DA9 previously blocked with casein/ 0.05% Tween 20/ 6% sorbitol/ 0.03% sodium azide, and developed in accordance with CDI protocol using europium-labeled mAb 77G7 and mAb RD3 for detection as described for mammalian prions [40,42,45,85,94]. The raw TRF signal was converted into the apparent $F_{app}$ as follows [40]: $F_{app}$ = ($TRF_{OBS}$ -$TRF_N$)/ ($TRF_U$ - $TRF_N$) where $TRF_{OBS}$ is the observed TRF value, and $TRF_N$ and $TRF_U$ are the TRF values for the native and unfolded forms, respectively, at a given Gdn HCl concentration [40,60,88]. To determine the concentration of Gdn HCl in which 50% of tau was unfolded ([Gdn HCl]$_{1/2}$), the data were fitted using the least squares method with a sigmoidal transition model (Eq 1):

$$F_{app} = F_0 + (F_{max} - F_0) / \{1 + e^{[(c_{1/2}-c)/r]}\}$$

(1)

The apparent $F_{app}$ in the TRF signal is a function of the Gdn HCl concentration(c), $c_{1/2}$ is the concentration of Gdn HCl at which 50% of the tau strains are dissociated/unfolded, and r is the slope constant [24]. Using these peak-derived Gdn HCl values, the CSA Fapp values at a given Gdn HCl concentration were compared in the individual rpAD and spAD cases using two-tailed ANOVA.

## Direct format of the CSA of tau conformers

Frozen aliquots of tau were thawed, sonicated for 3 × 5 s at 80% power with Sonicator 4000 (Qsonica), and 15 aliquots (15 µL) were treated with increasing concentrations of 8M Gdn HCl in 0.25 M or 0.5 M increments. After 30 min. incubation at room temperature, individual tubes were rapidly diluted with $H_2O$ containing diminishing concentrations of 8M Gdn HCl, so that the final concentration in all samples was 0.2 M. Each aliquot was immediately loaded in triplicates to dry white Lumitrac 600, High Binding Plates (E&K Scientific). Following incubation at 4°C and blocking with casein containing 0.05% Tween 20 and 6% sorbitol, plates were developed with europium-labeled mAbs 77G7 and RD3. The raw time-resolved fluorescence (TRF) signals obtained with the multi-mode microplate reader PHERAstar Plus (BMG LabTech) were converted into the apparent $F_{app}$ to obtain the concentration of Gdn HCl, where 50% of tau was unfolded ([Gdn HCl] 1/2), and the data were fitted using the least-squares method with a sigmoidal transition model as described for sandwich CSA [24].

## Enrichment of Sarkosyl-insoluble tau conformers for photochemical hydroxylation footprinting and mass spectrometry

The purification of insoluble tau conformers was adopted from previously described purification of different strains of human and animal prions with minor modifications [4,46,95]. Briefly, AD frontal brain cortex slices weighing ~10–25 g were disrupted to a final 5–20% (w/v) homogenate in ice-cold PBS (pH 7.4) containing 2% (w/v) Sarkosyl by three 50-s cycles in an Omni TH-01 homogenizer equipped with disposable plastic probes (Thermo Fisher Scientific, Waltham, Massachusetts, USA) and spun for 5 min. at 600 x g in Allegra centrifuge (Beckman Coulter, Brea, California) equipped with C0650 rotor to eliminate collagen and tissue fragments. The 5–20% brain homogenate containing protease inhibitor cocktail (0.5mM PMSF, and 2µg/ml of aprotinin and leupeptin, respectively) was clarified for 35 min. at 9500 rpm at 4˚C in Allegra centrifuge in C0650 rotor and the supernatant was spun for 30 min. at 185,000 x g at 4°C in Optima XPN-100 ultracentrifuge equipped with Ti 50.2 rotor (Beckman Coulter, Brea, California). The insoluble tau was resuspended and incubated overnight at 37°C with shaking at 600 rpm in 1.7M NaCl and 1% Sarkosyl in an Eppendorf Thermomixer (Eppendorf), and then centrifuged for 2 h at 26,000 × g and 10°C in an F2402H rotor of Beckman Allegra centrifuge. The partially purified samples containing insoluble tau were resuspended in 0.1% Sarkosyl containing 2mM $CaCl_2$ and incubated for 2 h at 37°C with 1,000 IU/ml of Collagenase (Worthington Biochemical Corporation) with shaking at 600 rpm. After centrifugation for 2 h at 26,000 × g at 10°C in an F2402H rotor of Beckman Allegra centrifuge, and incubation in 1.7M NaCl and 1% Sarkosyl overnight at 37°C followed by 2h spin as above, the pellets were then resuspended in PBS (pH 7.4) containing 0.1% Sarkosyl/ 5mM $MgCl_2$, and incubated with 50 IU/ml of Benzonase (Novagen) for 1 hr at 37°C. The final pellet obtained after final centrifugation at 26,000 × g for 30 min at 10°C in F2402H rotor of Beckman Allegra centrifuge was resuspended in 0.1% Sarkosyl containing protease inhibitor cocktail (0.05mM PMSF and 0.5 µg/ml of aprotinin and leupeptin, respectively), and stored at -80 °C for further analysis.

## Atomic force microscopy (AFM) imaging

Atomic force microscopy imaging was performed essentially as described previously [96]. In brief, tau aggregates purified from AD brain samples were deposited at room temperature on freshly cleaved mica discs. Following 1 min incubation (to adsorb the aggregates), discs were washed three times with Milli-Q water and then dried gently under the stream of nitrogen gas. Imaging was performed in a Scan Assist mode using a silicon probe (40 N/m spring constant) on MultiMode atomic force microscope equipped with a Nanoscope V controller (Bruker Instruments, Billerica MA). Images were processed using the Nanoscope Analysis software.

## Photochemical hydroxylation of tau proteins

Sarkosyl-insoluble tau was diluted 4-fold in ultrapure water to reduce the Sarkosyl content to 0.025% and spun down at 16,000 × g for 30 min. at 4°C in the S241.5 swing rotor of a Beckman Allegra centrifuge. Samples were resuspended in water with three 5s pulses of Sonicator 4000 (Qsonica) at 80% power, and aliquots containing 2–9 ug of protein per time point and 0.3% (v/v) $H_2O_2$ were pipetted into thin-walled PCR PR1MA tubes (MidSci). Photochemical hydroxylation was performed using a Spectrolinker XL-1000 (Thomas Scientific) equipped with 254 nm wavelength tubes at a continuous 3000–3800 uW/cm² energy output for 0, 2, 8, 16, 32, and 64 min. After irradiation, samples were immediately quenched with 20mM L-Met-NH₂ (L-Methionine amide hydrochloride, Sigma) and 0.1µM catalase (Sigma), and stored at -80°C until further testing. The unfolded monomers of His-tagged recombinant human 2N4R tau441 underwent the same hydroxylation protocol and were used as internal positive controls.

The photochemically hydroxylated and quenched samples were thawed and sonicated for 3 x 5s each, and an aliquot (10.5ul) at each time point was completely denatured using an equal volume of 96–98% formic acid (FA; Sigma)

and incubated for 30 min at room temperature. Following 50-fold dilution in ultrapure Milli-Q water, 200ul aliquots of each time point were loaded in duplicates onto a white Lumitrac 600 High Binding Plate (E&K Scientific) at the final 0.5-2.4 ug/ml of protein in 1% FA. The loaded plates were sealed, incubated overnight at 4°C in the dark, manually aspirated, and blocked with casein in 0.05% (v/v) Tween20 and 6% sorbitol (SurModics) for 1h at room temperature. Following one 350ul wash per well with 150mM NaCl containing 50mM Tris, 0.05% $NaN_3$, and 0.05% Tween20, pH 7.8, the tau in the plate was incubated with five Europium-conjugated anti-tau mAb's: A16040D (epitope 269–281; Bioleg-end), 1E1/A6 (epitope 275–291; Rohan de Silva), 77G7 (epitope 316–355; Biolegend), A16097B (epitope 359–373; Biolegend), and A16097F (epitope 368–441; Biolegend) diluted in Casein (Surmodics) containing 0.05% Tween20 and 0.05mM EDTA buffer as described in the CDI protocol. The Eu-N1 ITC (Perkin Elmer) labeling was performed as we described previously [45] with the following final Eu/IgG molar ratios: A16040D (12.8), 1E1/A6 (8.8), 77G7 (4.4), A16097B (4.1), and A16097F (2.7), respectively. Time-resolved fluorescence (TRF) signals of europium were mea-sured using a multimode microplate reader PHERAstar Plus (BMG LabTech). Recombinant monomers of 2N4R human tau441 expressed in E. coli with a His tag were used as positive controls. To compare the hydroxylation rates of anti-bodies with different absolute TRF (cpm) signals, the raw data were converted into the apparent fractional change $F_{app}$: $F_{app} = $ TRF(t)/ TRF($t_0$), where TRF$t_0$ is the observed TRF value at time 0 of hydroxylation. The data were fitted using the least-squares method with the Hill function (Eq 2) to determine the half-life ($t_{1/2}$) and hydroxylation rate ($r$) of each antibody epitope:

$$F_{app} = F_{base} + (F_{max} - F_{base})/[(1 + (t_{half}/t)^{rate}]$$

(2)

**Mass spectrometry (MS) analysis of the photochemical hydroxylation footprinting**

Before proteolysis, all samples were denatured and reduced by the addition of 96% formic acid (FA, Sigma) and 5mM tris 2-carboxyethylphosphine (TCEP, Sigma). The supernatant, dried overnight in a vacuum desiccator filled with solid $Na_2CO_3$, was solubilized in 25mM Ammonium bicarbonate (ABC) buffer, pH 8, conntaining 3.5M Urea and 5mM TCEP, sonicated, and mixed with Asp-N Sequencing Grade Protease (Aspartate-N, Promega) at 1:90 (w/w) enzyme/protein ratio. After 1 h of incubation at 37°C with shaking at 350 rpm, the reaction was continued for an additional 16h after adding recombinant trypsin (Roche) at a 1:90 (w/w) trypsin/tau ratio. The reaction was stopped by the addition of formic acid to a final concen-tration of 0.1%.

The identification and quantification of oxidative sites were performed by LC-MS analysis using an Orbitrap Eclipse mass spectrometer (Thermo Scientific, CA) interfaced with a Waters nanoAcquity UPLC system (Waters, MA, USA). Proteolytic peptides (~600 ng in 8µl; ~ 300ng in 4µl) were loaded onto a trap column (180 µm × 20 mm packed with C18 Symmetry, 5 µm, 100 Å; Waters) to desalt and concentrate the peptides. Peptide mixture was separated on a reverse phase column (75 µm x 250 mm column packed with C18 BEH130, 1.7 µm, 130 Å; Waters, MA) using a linear gradient of 0–32% mobile phase B (0.1% formic acid and acetonitrile) vs. mobile phase A (100% water/0.1% formic acid) for 60 minutes at 40°C at a flow rate of 300 nL/min. The eluted peptides were introduced into the nano-electrospray source at a capillary voltage of 2.0 kV. MS1 spectra were acquired for all eluted peptides in an Orbitrap mass analyzer ($R$ = 120 K: AGC target = 400,000; MaxIT = auto; RF Lens = 30%; mass range = 350–1500). MS/MS spectra were collected using a linear ion trap mass analyzer (rate turbo); AGC target, 10,000; MaxIT, 35 ms; $NCE_{CID}$, 35%). The resulting MS/MS spectra were searched against a database that consisted of a full tau protein sequence using the Mass Matrix software to identify specific sites of modification. In particular, MS/MS spectra were searched for peptides generated by digestion using mass accuracy values of 10 ppm and 0.8 Daltons for MS1 and MS/MS scans respectively, with allowed variable modifications for all known oxidative modifications previously documented for amino acid side chains [97,98]. Additionally, the MS/MS spectra for each site of the proposed modification were manually validated.

## Hydroxylation rate analysis

For the residue-level analysis, the integrated peak areas of the unmodified peptide ($A_u$) and each specific modified product ($A_m$) for this peptide derived from selected ion chromatograms were used to calculate the percent modification and fraction unmodified ($F_u$) for specific modification sites using the following equation:

$$Percent\ modification = [A_m/(A_u + \sum A_m)] \times 100$$

$$Fraction\ unmodified,\ F_u = 1 - [A_m/(A_u + \sum A_m)]$$

Where, $\sum A_m$ is the sum of all modified peak area from a particular peptide [46,99,100].

## Monitoring seeding of recombinant K18 with brain-derived tau

The expression of recombinant tau substrate and seeding reactions were monitored in real time-quaking-induced conversion (RT-QuIC) format as described previously [3,61,62,101] with the following minor modifications. The storage buffer of recombinant K18 was replaced with a 7 kDa cutoff Zeba Spin Desalting Column (Thermo Fisher) with 10 mM HEPES, 200 mM NaCl, pH 7.4. Buffer and protein concentrations were determined by measuring absorbance at 280 nm with a NanoDrop 1000 spectrophotometer and a molar extinction coefficient of 1490 M$^{-1}$ cm$^{-1}$. The seeding reaction buffer was prepared by diluting denatured and reduced K18 monomers to a final concentration of 5 µM with 10 mM HEPES, 200 mM NaCl, pH 7.4, and 10 µM thioflavin T (ThT) buffer containing 2.5 µM of low-molecular-weight heparin (MW ~ 4,350; Celsus Laboratories) in K18 reaction. AD brain homogenates were sonicated three-times for five seconds at 80% output (Misonix Sonicator 4000), serially diluted in TBS (pH 7.4), and 2µl of each dilution was delivered to four wells of a 96-well black optical bottom plate (Nunc) containing 98µl of seeding reaction buffer with substrate K18. The plates were covered with sealing tape (Axygen) and incubated at 37°C in a plate reader (BMG Labtech FLUOstar Omega) set to cycles of 60s double orbital mixing at 700 rpm for 60s incubation, with a one min pause to measure the ThT fluorescence. The kinetics of tau seeding and formation of misfolded tau aggregates was monitored in real time by the bottom reading of the fluorescence intensity every 45min using 450 ± 10nm excitation and 480 ± 10nm emission [60–62]. The raw fluorescence data were plotted as an average of four well readings, and the lag phase was assessed using an algorithm to detect the time and wells in which the ThT fluorescence exceeded by >6% of the background fluorescence of unseeded wells. The specific seeding activity is expressed as hrs of lag phase per ng of insoluble tau protein.

## AD-tau treatment of mouse primary neurons

Primary neurons were prepared at E16.5–17.5 wild-type C57Bl6/Tac mice as previously described [27,102]. All protocols were approved by the Institutional Animal Care and Use Committee of the Case Western Reserve University School of Medicine. Neurons were seeded on poly-D-lysine (PDL) pre-coated 96 well plates (Nunc MicroWell Manufacturer) at a density of 5 × 103 cells/well for confocal microscopy and on PDL-coated 12-well plates at a density of 2.7 × 105 cells/well for cell lysis. Every 5–6 days, 20% of the medium was removed and replenished with fresh complete neurobasal medium (supplemented with GlutaMAX and B27; Gibco) [103]. AD-tau inoculates were kept at −80°C, thawed on ice, and sonicated 3 × 5 s each at 80% power with a Sonicator 4000 before being added to the complete neurobasal medium. Primary neuronal cultures on day 7 (7DIV) were treated with AD-tau samples at a concentration of 45 ng/well in complete neurobasal medium in a 96-well plate and 180 ng/well in a 12-well plate format [27].

## Cell lyses of primary neurons

Cortical neurons in 12-well plates were washed twice with ice-cold PBS, lysed with ice-cold cell lysis buffer containing 1% Sarkosyl, a cocktail of protein proteases and phosphatases in PBS, and scraped (six wells for one treatment together in 300ul cell lyse buffer). Lysed cells were gently mechanically pressed through a syringe with a 25-gauge needle ten times, sonicated 3×5s each at 80% power with a Sonicator 4000, and pellets were collected at 14,000 rpm at 4°C for 30 min. using an Allegra X-22R tabletop centrifuge. The pellets were dissolved in ice-cold cell lysis buffer and centrifuged under the same conditions one more time. The supernatant was discarded, the pellets were dissolved in 150 μL of cell lysis buffer, and 50ul of samples collected from three independent experiments were analyzed immediately by sandwich-formatted CDI, as described previously [27].

## Immunocytochemistry with primary neurons

Treated primary hippocampal neurons were washed once with cold PBS and fixed at 21DIV with 100% ice-cold methanol for 15 min [27]. After three washes with PBS, cells were blocked with 10% normal goat serum containing 1% casein in PBS. The plates were incubated with primary antibodies (rat anti-mouse Tau clone RTM47 from FUJIFILM Wako, 1:1,000; mouse SMI312 from BioLegend, 1:500; rabbit MAP2 from SySy, 1:1,000) in PBS with 3% NGS overnight at 4°C. The wells were washed thrice for 5 min each with PBS, and secondary antibodies (anti-rat IgG/AF633; anti-mouse IgG/AF555, anti-rabbit IgG/AF488, all from Invitrogen, 1:500) were added for 1 h at 37°C. The cells were washed thrice for 5 min each in PBS and incubated with TrueBlack Plus Lipofuscin Autofluorescence Quencher in PBS (Biotium, 50ul/well) for 10 min. in the dark and washed thrice with PBS. Cells were mounted with Fluoromount-G containing DAPI (Invitrogen) and covered with a 5 mm cover glass (Electron Microscopy Sci).

## Culturing and differentiation of SH-SY5Y

Neuroblastoma SH-SY5Y cells were purchased from the American Type Culture Collection (ATCC number CRL-2266) and maintained in standard culture condition - Dulbecco's Modified Eagle Medium/Nutrient Mixture F-12 with GlutaMAX supplement (Gibco) supplemented with 10% fetal bovine serum (FBS, Sigma) and Penicillin-Streptomycin mixture (10 μl/ml, Sigma) at T75 flask in 5% CO2 and 37°C. All cells used for the experiments were maintained under passage 33 (cells were purchased at passage 28). Cells were seeded in dishes pre-coated with GFR Matrigel (Corning, 8.7 ug/cm2 density) according to manufacturer instructions and Matrigel was also added to cell suspension (40 μg/ml) immediately before seeding. The cells were cultured at a density of 6.5×103 cells/well in a 96-well plate (Cellvis) for immunofluorescent staining and LDH assay, 2.5×105 cells/well in a 6-well plate (VWR) for western blots, and 1.6×104 cells/well in a dish (10 mm glass diameter, MatTek) for calcium imaging and STED imaging. Twenty-four hours after seeding, FBS levels in the medium were reduced to 5% and all-trans retinoic acid (RA) was added to a final concentration of 10 μM. Three days later, the medium was fully exchanged for medium with 2% FBS and 10 μM RA for another three days.

## Treatment of differentiated SH-SY5Y

Cells were inoculated with AD-tau PTA-enriched samples before adding medium with 2% FBS and 10 μM RA. Briefly, AD-tau fractions were mixed with Lipofectamine 3000 transfection reagent (Invitrogen) for 30 min at RT in Opti-MEM reduced serum medium (Gibco) and added to cells in concentrations 30ng/50μl for a 96-well plate, 60ng/80μl for MatTek dishes, and 300ng/1.5ml. After 6 hours in 5% CO2 and 37°C, cells were washed with pre-warmed PBS and DMEM/F12 medium supplemented with GlutaMAX, 2% FBS, and 10 μM RA was added for 1 hour and 72 hours. A control sample extracted from brain homogenate with other neurological diseases (OND) was added in a volume corresponding to the AD-tau sample containing the lowest levels of sarkosyl-insoluble tau.

## LDH assay

LDH-Glo Cytotoxicity Assay (J2380, Promega) was applied to measure the amount of lactate dehydrogenase (LDH) released to media [27]. The media from all wells were collected after 72 hours of treatment and stored in LDH storage buffer (200 mM Tris–HCl pH 7.4, 10% glycerol, 1% BSA) at a ratio of 1:5 at − 20 °C till use. After calibrating samples to room temperature, 12.5 µl of samples were transferred into light grey half-area Alpha plate-96 (Costar) and followed by 12.5µl of LDH detection enzyme mix with reductase substrate. Every plate contained a calibration LDH curve as a control of linear range. The bioluminescence was measured by PHERAstar after 30 min incubation at RT. Data were collected from three experiments and analyzed with one-way ANOVA/Multiple comparisons.

## Immunocytochemistry with SH-SY5Y

The cells were washed once with cold PBS and fixed with a mixture of 4% PFA/0.2% glutaraldehyde (Sigma) or 4% PFA with 1% Triton-X100 on an ice block for 15 min. After three wash steps with PBS, PFA-fixed cells were permeabilized with 0.1% Triton in DPBS and blocked with 10% normal goat serum (NGS, ThermoFisher Scientific) and 1% casein in PBS. The plates were incubated with primary antibodies in PBS-T with 3% NGS overnight at 4 °C. Applied primary antibodies were rat monoclonal anti-human tau (1:1000, clone RTM49, Wako, for STED 1:400), chicken polyclonal anti-β3-tubulin (Tuj1, 1:1000, Synaptic Systems), mouse monoclonal anti-human PHF-Tau (1:1000, clone AT8, pS202/pT205, ThermoFisher Scientific), rat monoclonal anti-tau (1:2500, clone A16040D, Biolegend), polyclonal chicken anti-MAP2 (1:1000, Synaptic Systems), polyclonal rabbit anti-PSD95 (1:500, Invitrogen, for STED 1:200), mouse monoclonal anti-synaptophysin 1/2 (1:250, Synaptic Systems). Wells were washed 5 min three times with PBS-T and secondary antibodies (Invitrogen, for STED Abberior) and Phalloidin-TRITC (1:250, Sigma) diluted in PBS-T with 3% NGS were added for 1 h at 37 °C. Cells were washed for 5 min three times with PBS, mounted with Fluoromount-G with DAPI (Invitrogen), and covered with 5 mm cover glass (Electron Microscopy Sci). Confocal images were acquired with Leica HyVolution SP8 confocal microscope, objective 40x/oil with z-stacks of 0.40µm, and three independent experiments were performed. STED images were acquired with Leica TCS SP8 gated STED 3X, objective 100x/1.4 (oil), and deconvolution followed in Hyugens software. The images are presented as max intensities restored from 15 z-stacks in ImageJ. The signal quantification corresponding to triton-insoluble tau was performed from max-intensity image projections from three independent experiments with multiple wells. The threshold (0–40) binary images were applied to define signal-positive regions of interest (ROIs) to calculate intensity density (IntDen) in channels corresponding to the tau fluorescent signal. We expressed the data as an x-fold increase of tau signal to untreated cells (medium only), mean ± SEM, and an unpaired t-test was performed.

## Cell lysis of treated SH-SY5Y and western blots

Treated RA-differentiated SH-SY5Y in 6-well plate were washed twice with ice-cold PBS, lysed with ice-cold cell lyse buffer containing 1% Triton, a cocktail of protein proteases and phosphatases, benzonate (50 IU, Sigma), and 5 mM MgCl2 in PBS, and scraped (three wells for one treatment together in 500µl cell lyse buffer). Lysed cells were kept 20 min on ice, sonicated 3 × 5 s each at 80% power with Sonicator 4000. The samples were gently mechanically pressed through a syringe with a 25-gauge needle ten times and kept another 20 min on ice on a shaker (300rpm). Pellets were collected at 14,000 rpm at 4 °C for 30 min in Allegra X-22R tabletop centrifuge. Supernatants were transferred and stored at −80 °C till further applications. Pellets were dissolved in 30 µl of 1x Licor Laemmli buffer with β-mercaptoethanol (BME) and 20 mM Tris-HCl pH 6.8 and sonicate 3x 5 s each at 80% power. Supernatants were mixed with 4x Licor Laemmli buffer with BME and 20 mM Tris-HCl pH 6.8. All samples were heated at 98 °C for 5 min. The western blots were performed as described previously [16]. The PVDF membranes were blocked with Intercept blocking buffer (TBS, Licor) for 1 h at RT and incubated with primary antibodies (anti-tau, clone A16040D, dilution 1:2000) overnight at 4 °C. After washing four times for 5 min with TBS-T, incubation with secondary antibodies (IRDye 680RD Goat anti-Rat IgG, Licor, dilution 1:15000) followed

for 1 h at RT. After four wash steps for 5 min each with TBS-T, the IR detection was performed with the Odyssey Infrared Imaging System (LI-COR Biosciences).

## Calcium imaging

Changes in intracellular calcium were evaluated using the calcium-dependent fluorescent dye Fluo-8 (Life Technologies, AAT Bioquest). RA-differentiated SH-SY5Y cells inoculated with AD brain-derived tau for 3 days were washed with DPBS and incubated with Fluo-8 mixed with Pluronic F-127 (Biotium) in a 1:1 ratio to a final concentration of 1 mM Fluo-8 in modified Krebs buffer (20 mM HEPES pH 7.4, 130 mM NaCl, 4.7 mM KCl, 1.3 mM CaCl2, 1 mM MgCl2, 1.2 mM KH2PO4, 5 mM glucose) with Hoechst 33342 solution (BD Pharmingen) for 40–60 min at 5% CO2, 37 °C. Cells were washed twice with a Ca2+-free HBSS medium and left in a Ca2+-free HBSS medium to equilibrate for 5 min. Fluorescence was measured by Leica HyVolution SP8 confocal microscope, 40x/1.30 (oil immersion). First, a single image of the Hoechst signal was acquired (excitation at 405 nm). Cells were excited at 488 nm wavelength and images were acquired at 500–520 nm in the xyt mode (2 s frame) for 3 min (90 frames) as a baseline signal (the resting fluorescence before stimulation, F0) and 7 min (210 frames) after the addition of KCl to a final concentration of 50 mM (F). Fluorescence signals were expressed as a ratio ($\Delta F/F0$) of the mean change in fluorescence ($\Delta F$) relative to the resting fluorescence before stimulation (F0) using ImageJ. Fluorescence intensities were evaluated in individual cells defined by single-cell regions of interest (ROI) identified based on images of the Hoechst signal. Data are given as geometrical mean ± SEM from three independent experiments. The intensity of the peak fluorescence signal was measured at 30 s after 50 mM KCl stimulation (t = 210 s).

## Statistical analysis

Tau protein data and clinicopathological profiles were investigated using our previously published analysis of variance (ANOVA) protocols [3,46]. To test the effect of different epitopes of brain-derived tau on seeding potency, we applied linear and nonlinear regression models using Igor Pro 9 (WaveMetrics) and SPSS 28 (Statistical Package for Social Sciences, SPSS Inc.). Two-sided power analysis to determine the impact of sample size and all other statistical analyses were performed using the SPSS 28 software. All cell experimental data were normalized with respect to medium controls and represented as mean ± standard error of the mean. Statistical evaluation off cell data was done using GraphPad Prism 6 software, KaleidaGraph, and Excel.

## Supporting information

**S1 Fig. Differential decay of antibody epitopes induced by photochemical hydroxylation and mass spectrometry analysis of representative AD brain-derived tau samples and rectau protein.** (**A**) Sensitivity and dynamic range of five antibodies in detection of monomeric recombinant tau441. (**B**) Normalized hydroxylation decay curve for R1 (269–281) domain, (**C**) R2/3 (275–291) domain, (**D**) R3/4 (316–355) domain, (**E**) R4 (359–373) domain, (**F**) C-terminal tail (368–441) domain, and (**G**) percentage of hydroxylated residues after 8 min photochemical hydroxylation analyzed by mass spectrometry with two typical AD cases and recombinant tau441. The boxes and lines in the domain plots individual data values of apparent fractional change ($F_{app}$) of each sample in the transition from not hydroxylated state (time 0) to hydroxylated state obtained from duplicate measurements. The curves in plot **B** through **F** are the least square fits of the data with Hill model.
(EPS)

**S2 Fig. Detergent-insoluble AD brain-derived tau conformers used in the hydroxylation and seeding experiments.** (A,B,C,D) Representative AFM images of different types of tau aggregates employed in this study. (A,B) Different magnifications of sarkosyl-insoluble fraction (P5) obtained after high-speed centrifugation step of AD brain homogenate; (C, D)

final centrifugation pellet (P7) from AD brain used in hydroxylation and seeding experiments. Colored bars at the right side of each image represent a height scale. (E,F) Conformational stability assay (CSA) profiles of Sarkosyl-insoluble tau in brain homogenates and enriched by high-speed centrifugation in two representative AD cases. The boxes and lines in the plots are mean ± SEM values of apparent fractional change ($F_{app}$) of each brain sample in the transition from native to denatured state obtained from triplicate measurements at each concentration of denaturant (Gdn HCl) with Europium-labeled 77G7 monoclonal antibody. The data are fitted with sigmoidal transition model to obtain the 50% of unfolded tautau expressed as midpoints ($x_{half}$) of CSA $F_{app}$, $[GdnHCl]_{1/2}$ ± S.E.M. (G) Representative silver staining and WBs of serial dilutions of AD brain-derived tau (P7) used in hydroxylation and seeding experiments. WBs were performed as described previously [3] and developed with mAb 77G7 for total (3+4R) tau and mAb RD3 for specific detection of R3 isoform.
(EPS)

**S3 Fig. Photochemical hydroxylation-induced decay of antibody epitopes monitored in representative (A) AD brain-derived Sarkosyl-insoluble tau and (B) recombinant tau441.** The WBs were developed with antibody 16040D for R1 (269–281) domain; RD4 for R2/3 (275–291) domain; 77G7 for R3/4 (316–355) domain, and 16097B for R4 (359–373) domain.
(EPS)

**S4 Fig. Hydroxylation footprinting of different MTBD epitopes monitored with Europium-labeled antibodies.** (**A**) Normalized hydroxylation decay curve for R1 (269–281) domain, (**B**) R2/3 (275–291) domain, (**C**) R3/4 (316–355) domain, (**D**) R4 (359–373) domain, (**E**) C-terminal tail (368–441) domain, and (**F**) comparison of different domains averaged for each domain. Each decay curve was generated from the Sarkosyl-insoluble tau enriched from frontal cortex of one case of AD (n=22). The lines and shades in the plots are mean ± S.E.M. values of apparent fractional change ($F_{app}$) of each brain sample in the transition from non-hydroxylated state (time 0) to hydroxylated state obtained from duplicate measurements at each timepoint.
(EPS)

**S5 Fig. Differentiation of SH-SY5Y neuroblastoma line into neuronal-like cells. (A)** The RA-differentiated SH-SY5Y cells form projections connecting adjacent cells visible by phase-contrast. Representative confocal images showing markers typical for neuronal cells are present, Tuj1 (cyan), tau (red), and MAP2 (blue), and resembling the immunostaining patterns of polarized axon-dendritic morphology. Scale bar is 2 µm for phase-contrast image and 50 µm for confocal images. **(B)** Confocal images showing the cellular morphology after immunostaining with actin (grey) and DAPI for nuclei (blue). Differentiated SH-SY5Y cells expressed pre- and post-synaptic markers (synaptophysin – green, PSD95 – magenta) also localized in forming cellular projections. Scale bars of 50 µm and 20 µm for cropped images. **(C)** Ionomycin-induced calcium influx in differentiated SH-SY5Y measured in 2 s intervals with laser scanning confocal microscope as 3 min interval as a baseline Fluo-8 fluorescence and 7 min interval after calcium influx in presence of 1 µM ionomycin. **(D)** Representative confocal images of baseline Fluo-8 immunofluorescence at 3 min time point and in the fluorescence peak at 3.5 min, 30 s after ionomycin exposure.
(EPS)

**S6 Fig. Human tau misfolding and calcium influx changes in differentiated SH-SY5Y cells inoculated with AD brain-derived tau. (A)** Fluo-8 fluorescence expressed as $\Delta F/F_0$ for controls (medium and OND) and cultures inoculated with AD brain-derived tau (n=6) for 3 days. We captured confocal time-lapse images: 90 frames for baseline Fluo-8 fluorescence in 2 s intervals and 210 frames after addition of KCl in 2 s intervals. Each individual curve represents a mean of geometrical means of signals from individual cells as ROIs determined based on Hoechst 33342 signal from three independent measurements. **(B)** Fluo-8 fluorescence intensity in ratio to baseline fluorescence at max peak (t=210 s). Each bar encloses the data with mean value ± SEM of geometrical means of individual cell signals in three independent

experiments, numbers of total cells are marked in boxes. Significance was determined with student t-test (P-values). **(C)** Accumulation of human tau (antibody 16040D signal, green) and hyperphosphorylated tau (antibody AT8 signal, red) expressed as a ratio of Int. Density fluorescence signals in SH-SY5Y cells at three days and one hr after inoculation with AD tau. Each bar encloses the data with mean ± SEM from three independent experiments. **(D)** Regression analysis of relationship between Ca influx and accumulation of misfolded human tau protein (A16040D) and **(E)** hyperphosphorylated tau protein aggregates (AT8) induced by different isolates of AD brain-derived tau protein (n = 6). Each data point represents separate experiment, NS – not significant.
(EPS)

**S1 Table.  Case information for all samples used in the study.** Demographics of AD cases, disease duration (DD), neuropathological classification of AD stage, postmortem intervals (PMI), ApoE alleles, and conformation dependent assay (CDI) data on Amyloid beta 42 (Ab42) and detergent-insoluble TAU in the cortex.
(DOCX)

**S2 Table.  Conformational stability assay (CSA) results on brain-derived, detergent-insoluble tau aggregates in all AD cases used in the study.**
(DOCX)

**S3 Table.  Hydroxylation half-life of different microtubule binding domains (MTBDs) and C-terminal tail monitored in insoluble AD brain-derived tau conformers in all AD cases used in the study.**
(DOCX)

**S4 Table.  Summary data on primary mouse neuron cultures and neuronally differentiated SH-SY5Y cells exposed to AD brain-derived tau conformers.**
(DOCX)

## Acknowledgments

The authors are grateful to the patients' families, the CJD Foundation, referring physicians, and all members of the CWRU Alzheimer's Disease Center and National Prion Disease Pathology Surveillance Center for their technical help and review of the data. We are grateful to Alexander Miron for Illumina and Sanger sequencing and we are indebted to all brain donors and their families for generous brain donation for research.

## Author contributions

**Conceptualization:** Jiri G. Safar.

**Data curation:** Lenka Hromadkova, Chae Kim, Mohammad Khursheed Siddiqi, Janna Kiselar, Jiri G. Safar.

**Formal analysis:** Chae Kim, Jiri G. Safar.

**Funding acquisition:** Lenka Hromadkova, Jiri G. Safar.

**Investigation:** Lenka Hromadkova, Chae Kim, Tracy Haldiman, Mohammad Khursheed Siddiqi, Krystyna Surewicz, Kiley Urquhart, Dur-E-Nayab Sadruddin, Lihua Peng, Xiongwei Zhu, Witold K Surewicz, Mark L. Cohen, Rohan de Silva, Janna Kiselar.

**Methodology:** Lenka Hromadkova, Chae Kim, Tracy Haldiman, Mohammad Khursheed Siddiqi, Krystyna Surewicz, Kiley Urquhart, Dur-E-Nayab Sadruddin, Mark R. Chance, Janna Kiselar, Jiri G. Safar.

**Project administration:** Jiri G. Safar.

**Writing – original draft:** Jiri G. Safar.

**Writing – review & editing:** Lenka Hromadkova, Chae Kim, Tracy Haldiman, Krystyna Surewicz, Dur-E-Nayab Sadruddin, Lihua Peng, Xiongwei Zhu, Witold K Surewicz, Mark L. Cohen, Mark R. Chance, Rohan de Silva, Janna Kiselar, Jiri G. Safar.

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
