## [Decision Letter · Decision Letter 0]

PPATHOGENS-D-25-00167

Structural exposure of different microtubule binding domains determines the propagation and toxicity of pathogenic tau conformers in Alzheimer’s disease.

PLOS Pathogens

Dear Dr. Safar,

Thank you for submitting your manuscript to PLOS Pathogens. After careful consideration, we feel that it has merit but does not fully meet PLOS Pathogens's publication criteria as it currently stands. Therefore, we invite you to submit a revised version of the manuscript that addresses the points raised during the review process. While the reviewers agreed in general that the paper addressed an important issue that would be of interest to other researchers, they raised several important points with regard to the methods and subsequent data interpretation that need to be addressed. In particular, please be sure to directly address the following issues: 1) the use of age matched, non-Alzheimer's disease brain and soluble tau as controls, 2) the possibility that the differences observed were due to associated factors bound to tau and not differences in tau conformation, 3) experimental variability in the guanidine denaturation assay and, 4) the clarity and interpretation of the data presented in multiple figures as specified by Reviewers 2 and 3.

Please submit your revised manuscript within 60 days Apr 27 2025 11:59PM. If you will need more time than this to complete your revisions, please reply to this message or contact the journal office at plospathogens@plos.org. Please include the following items when submitting your revised manuscript:

We look forward to receiving your revised manuscript.

Kind regards,

Suzette A. Priola

Academic Editor

PLOS Pathogens

Surachai Supattapone

Section Editor

PLOS PathogensSumita Bhaduri-McIntosh

Editor-in-Chief

PLOS Pathogens

orcid.org/0000-0003-2946-9497

 Michael Malim

Editor-in-Chief

PLOS Pathogens

orcid.org/0000-0002-7699-2064

**Additional Editor Comments (if provided):**

The reviewers agreed in general that the paper addressed an important issue that would be of interest to other researchers. However, one reviewer was concerned that, although heterogeneity in tau assembly structures and seeding activity was observed in Alzheimer's disease cases with distinct phenotypes, these differences were associational and not correlative. All of the reviewers expressed some concerns about the approach used including the appropriateness of some of the controls, the possibility that the differences observed were due to associated factors bound to tau and not differences in tau conformation, the clarity and interpretation of the data presented in multiple figures, and the validation of some of the methods, particularly with regard to defining experimental variability in the guanidine denaturation assay. Given that addressing these issues will likely require new experiments, the decision of Major Revision was made.

**Journal Requirements:**

**Reviewers' Comments:**

Reviewer's Responses to Questions

**Part I - Summary**

Reviewer #1: Hromadkova and colleagues describe a series of experiments (in vitro and cell culture) which implicate specific MTBD domains in disease progression and cognitive decline. Using a novel Eu-mAb labeling approach, CSA and RT-QuIC assays the authors identify R4 exposure as the most variable domain between patients. Additionally, neuronal cell culture was imperative in identifying the R1 domain as the source of cytotoxic effects via calcium influx. These findings are exceptionally interesting, and very important in the larger discussion on tau templating and the basis of disease variation between patients. This paper is clearly written and is a nice example of the relationship between fundamental protein structure and ‘real-world’ disease implications.

Reviewer #2: Cryo-EM studies have suggested that tau assemblies in AD are relatively uniform in structure. Yet there is considerable diversity in progression rate and phenotype. The authors hypothesize that this could be due to variation in tau assembly conformation. This work is thus tackling an important question. It suggests that the cryoEM story of uniform tau structures in disease states may not be completely true. Consequently, this study seeks to investigate diversity of tau seed structures in different forms of AD using epitope availability assays and seeding activity in cell models. The general question: to what extent do differences in tau assembly conformation (presumably in subsets of tau assemblies) account for clinical variation in patients? The methods as applied, which are based on modification of solvent-exposed regions and determination of the effects on epitope accessibility, however, leave considerable uncertainty. Additionally, the parsing of AD cases leaves much to be desired. Simply dividing cases as “rapid” vs. “slow” progression is unlikely to reveal much, as these can be very subjective determinations. To effectively challenge this model of uniformity of tau core assembly structure, the authors must do a better job of defining the seed-competent species they are studying. Right now the measures are associational, the correlations are not robust, and it is difficult to draw firm conclusions from the work.

Reviewer #3: The paper by Hromadkova is a valiant effort to better understand the relationship between the properties of tau aggregates and the progression of Alzheimer's disease. While the work may shed light on the fundamental mechanics of Alzheimer's disease and may inspire unique diagnostic tools, more work is needed before publication.

Novelty-The authors adapted a hydroxyl damage and antibody based amino acid modification technique to the study of Alzheimer's disease. The technique has the potential to be a sensitive tool for distinguishing variability in tau structure.

Strengths/weaknesses - It is evident that a lot of work has been put into the validation and experimentation in this work, however more work must be done to properly validate the methods and ideas presented in this paper. One of the greatest strengths is the usage of aggregated tau form human patients while one of the greatest weaknesses is the poor presentation of data in the figures.

General execution - The text is well written and clear but the Figures muddle the entire work with inconsistencies, poor layout, and low quality presentation of data.

**Part II – Major Issues: Key Experiments Required for Acceptance**

Reviewer #1: 1. Why was recombinant tau (rectau441) used in the hydroxylation decay experiments as opposed to a control (healthy) brain homogenate? Control brains of age matched patients likely contain tau aggregates that never progress to AD and would perhaps produce a more suitable comparison.

2. For the RT-QuIC assays, there is no mention of a reducing agent in the RT-QuIC buffer, how is K18 reduced? Also, in most published tau RT-QuIC assays seeding in control brains has been demonstrated. Does the K18 assay as presented in this work show any seeding activity of control brains at 10-4? Furthermore, a full dilution series for at least one AD brain should be presented to indicate that the seed is being diluted out.

3. It is fascinating that R4 was most variably protected (359-373), and the C-tail (368-441) was the least despite the slight overlap in sequence. Does the Fitzpatrick cryo-EM structure of AD or any of the Scheres cryo-EM structures provide any insight as to why these residues may be important in driving seed potency?

Reviewer #2: Major Points

1. The authors appear to infer causality from association. That is, they find heterogeneity in tau assembly structures and seeding activity associated (but not overwhelmingly so) with brain homogenates from AD cases of distinct phenotype. On its own these measures are associational, but don’t reflect causality.

2. Assumptions about structure/conformation are based on solvent accessibility of specific domains. This method is generally acceptable in the setting of purified forms of protein, where it is unlikely that other factors are not present in the complexes studied. In this case, however, the authors are studying tau within brain homogenates. How can the exclude the possibility that any differences they see are not due to associated factors bound to and masking components of otherwise identical assemblies? This could easily confound their data.

3. Orthogonal approaches to study structures are needed. Cryo-EM (even 2D reconstructions) or some orthogonal approach to compare seed structure (e.g. recently published Vaquer-Alicea et al. Sci Adv. 2025) would help solidify their findings. Otherwise, we are left with limitations raised in point (2) above.

4. Resolve patterns within defined assemblies. Failing orthogonal validation (much preferred), it would also be useful to extract soluble tau, resolve the oligomers by SEC (or even look at soluble tau overall) and perform the OH labeling. By restricting their analyses to insoluble tau they are potentially biasing the analyses.

5. On its own, Figure 2 is difficult to interpret in terms of significance. Data underlying the figure should be clearly shown. It appears data derive from one case (according to figure legend). How is this significant?

6. Cell death data in Fig. 4b is weak and unconvincing. This type of assay can’t reliably be used to indicate differences in tau assembly conformation, as it is going to be highly susceptible to artifacts based on differences in disease state between individuals (e.g. degree of insoluble material).

7. Unclear what is the point overall of Figs. 4, 5. It is well known that AD lysates have more seeding activity vs. controls. Therefore it is unclear why these figures are included, and how they advance the thesis of the paper.

8. Variance in assays must be established. The authors make a great deal of variance in Gdn denaturation (Fig. 1), but it is unclear what is the variance in this assay with samples of relatively similar pathology. That is, just because there is variance in the assay, it doesn’t mean that there are different conformations in the cases—the assay itself may be quite variable. They must provide “calibration” cases to show the denaturation curves are meaningful.

Reviewer #3: Text - In general the text reads well and brings up intriguing points. However, the paper needs more controls in critical areas to support the authors claims.

Major Points:

1. The authors report that “The enrichment protocol for AD brain-derived insoluble tau was adopted from the purification of distinct strains of prions without proteases [36,38] and was designed to maintain full-length tau and remove noncovalent interactors”, however there is no data showing that noncovalent interactors have been removed. For the prion protein proteases like PK remove many of the non-protease resistant proteins helping to remove non-prion protein, however, even with multiple rounds of protease treatment, PTA precipitation, and resuspension small amounts of non-prion proteins, or their degradations products, persist. The authors should show blots that indicate that non-covalent interactors have been removed.

2. The authors show in supplemental Figure 3 that the amount of tau detectable by western blot decreases over time, which is equated to the loss of the epitope to which the antibody binds. However, the isolated tau in the blots is much larger than the purified tau, indicating that the material on the blots is not denatured fully and retains its aggregated structure while being run over a gel. If preparation for blots is unable to break up the larger aggregates of tau then it is possible that the apparent loss of tau on the blots may do not be due to hydroxyl damage to antibody epitopes and maybe the result of changes in the aggregate structure that exclude antibody interaction. Another possibility is that hydroxyl treatment induces further aggregation of tau to the point that it can no longer enter the gel. The authors should demonstrate that the change in tau signal in the western blots is not the result of more aggregation with a sedimentation assay to evaluate aggregation state, a dot blot that captures aggregates off all sizes, or by showing a Coomassie stain or silver stain of the gels associated with these blots, which would have to be a repeat of the experiment unless the authors have this data on hand.

3. It is difficult to evaluate the calcium signaling experiments with the Figure being incomplete.

4. In supplemental Figure 1 G the region encompassing R4, 337 to 369, shows similar hydroxylation between AD1, AD2, and recombinant Tau, though it is difficult to make statistical arguments with only duplicated samples. With all samples showing similar hydroxylation is hydroxylation at R4 actually a good site for monitoring structural/conformational properties in tau that dictate conversion rates? The authors should comment on this aspect of their data.

Figures - While the text is well written and to the point, the figures need an overhaul. As they currently stand the figures are inconsistent and difficult to follow.

Major Points:

1. Figure 5 panel F is missing.

2. Figure 5B is missing labels on the x-axis and the line for the y-axis. The y-axis is also missing its label.

3. Supplemental Figure 3 appears to be pixelated, and authors should check the dpi of gel images.

4. Text on images in Figure 3D is pixilated and should be converted into a vector formation. The image dpi appears to be fine but the distortions on the text make it difficult to read.

5. Text on images in Figure 4F is pixilated and should be converted into a vector formation. The images also appear a bit fuzzy and readers would benefit form having higher quality images to look at.

6. Error bars are cut off in supplemental Figure 2.

7. It is not visually clear what the significance indicator “*” in Figure 4B refers to. Please consider using lines to make it visually more clear which data sets are significantly different as was done in panel E.

8. Many Figures use data without labels and this creates confusion. Figure 3A has many lines with different colors and it is not clear what each line goes to. If it is simply showing the range seeding times then that should be made clear, however, the Figure legend and text give the impression that these lines may show both test samples and controls. Differences in samples should be indicted with clear color distinctions.

9. Figure 3D has a critical scaling issue. Does the dashed line box inset in the larger image represent the lower image exactly? If so there is a problem with the scale bars. The smaller box is roughly one third of the width of the image, yet the scale bar is only different by about 1.25 x.

**Part III – Minor Issues: Editorial and Data Presentation Modifications**

Reviewer #1: 1) Abstract, last sentence change “ti”, to “it”.

Reviewer #2: (No Response)

Reviewer #3: Text - While many of the minor points listed below are easily fixable with changes in wording to the text, some of the points may be more critical points depending upon how the authors address them.

Minor Points:

1. Figure legends use capitalization and acronyms inconsistently. Figure 3 legend both “thioflavin” and “Thioflavin” in the same sentence.

2. Spacing appears inconsistent in reference brackets. See “K18 (4R) tau substrates [44,45 ] (Fig. 3a” and compare to “AD cases with faster clinical progression [3]”

3. The authors report “SDS-insoluble high-mass oligomers of tau and a limited ~20-fold dynamic range precluded quantitative evaluation (Fig. S3)”. Is the material shown in Figure S3 the insoluble mass? If so, is there material that was not able to make it into gels because it could not be denatured at all in sample buffer?

4. The authors indicate that the R4 domain is the most “reliable predictor of seeding activity” as the other domains show poor correlations. However, the R value is ~0.39 for R4, which is not a very impressive R value despite the significance of the fit being 0.0003. The authors should comment on this discrepancy and what the low R value means for the interpretation of their data / models.

5. The difference in the R and P values for R4 and R3/4 are minimal. What is the cutoff that justifies R4 being acceptable to use but not R3/4?

6. The authors should add speculation on why the other domains tested have such a high variability compared to R4.

7. Supplemental Figure 1 G is difficult to read and it is hard to compare stretches of amino acids due to the way data is shown. This Figure is a critical support for their work that should be reworked to make it easier to interpret.

8. The data in supplemental Figure 1 G takes and SEM of duplicate samples. Error analysis on duplicated samples is controversial and Figures should show individual points with trends or add a third sample.

Figures

Minor Points:

1. Graphics in Figures 1 and 4 appear pixelated and should be recreated with a higher dpi image or replaced with vector graphics.

2. Text overlaps in non-conventional ways in Figure 5 A. The “/” overlaps with the “)”. This is likely a formatting / graphical issue and should be fixed.

3. Text in Figures have inconsistent spacing. To be specific single and double spacing are used inconsistently.

4. The text in supplemental Figure 3 that sits on gel images appears stretched.

5. Supplemental Figure 5C appears to be a pixilated image as opposed to a vector graphic. Please either upload a higher quality image or convert to a vector graphic.

6. Spacing of panels is inconsistent between Figures and the content of individual Figure sections often overlap into other figure sections creating confusing layouts that are hard to follow. Please clean up figure spacing.

7. Some images are too dark to properly see immunofluorescence staining and should be brightened or the image quality should be enhanced.

8. Figure 2 inconsistently uses white boxes behind text. Numbers for half life have white boxes behind them which makes them easy to ready but other numbers do not, “269-281”. The text without boxes are hard to read and the cut into Figure graphics altering their appearance. Data in plots also cuts through numbers making it hard to read the numbers. In the upper panel, orange data lines, the number “10” is partially cut off by an invisible box and the text box behind the number has white borders that create a distortion on the plot. Plot layout, numbers, and data graphics overlap in inconsistent ways with some obscuring lines and numbers. This should be fixed.

9. The usage of different colored text to indicate different proteins, such as TuJ1 and Tau, is a great idea but the color scheme used will make it difficult to distinguish for some readers. TuJ1 and Actin text color in supplemental Figure 5 were particularly difficult to discern for me.

10. Through Figures spacing between numbers and units is inconsistent. Supplemental Figure 6 uses “50mM” with no space and Figure 5 uses “30 ng” with a space. Please make spacing consistent throughout the Figures.

11. Text color and font size are inconsistent through the Figures. Some text appears black while some text nearby appears a dark grey.

12. Supplemental Figure 6 is missing “(“ from y-axis labels for panels E and D.

13. Error bar presentation is not consistent across Figures or within them. In supplemental Figure 6 the lower half of error bars are present in panel B but not in Panel C.

14. Figure 4B, the colors in the bars bleed outside the lines in inconsistent ways.

15. Capitalization and punctuations are used inconsistently throughout Figures. For example, the word “density” is lower case on the y-axis of 4E but uppercase on the y-axis of 5C. Also, in 5C “Dif” appears with a “.” while “Int” in 4E does not. Please correct consistency in capitalization and punctuation.

16. Nomenclature varies within Figures. In supplemental Figure 6 panel B y-axis reads “(peak amplitute at 21 0 s)” while Panel D reads “Peak Apmplitude, t = 210s)”. It would make it easier for readers to interpret the paper if nomenclature was similar. Try making titles consistent when referring to similar things by using “at 210s” or “t=210s” instead of mixing within the same Figure.

17. In supplemental Figure 6D “amplitude” is mis-spelled as “Apmplitude”. In the same Figure panel C “Accumulation” is mis-spelled as “Accumulationn”. In the same Figure panel B “amplitude” is mis-spelled as “amplitude”. Please check for spelling errors in the Figures.

18. Usage of parenthesis and brackets is inconsistent throughout the Figures and within the same Figure. For example, supplemental Figure 5D uses “(3 min)” and “[3.5 min]” right next to each other. If the parenthesis and brackets do not represent something in specific please chance Figures to use one or the other consistently.

19. Font usage is inconsistent throughout the Figures and within Figures. In supplemental Figure 6 the delta symbol has as different font in panel E and A.

20. The symbol for hydroxyl groups in the Figures is not used consistently. In Figure 1C it appears as “•OH” with a circle in front of it, in supplemental Figure 1B it appears as “OH” with no symbol, and in supplemental Figure 3 it appears as with a square symbol in front of it.

21. Size bars are not consistently indicated in Figures. Figure 5 has no numbers or units denoting the size of the bars in the image while Figure 4 does. Also, the text on some of the bars in Figure 4F are too small to read. They should be replaced with vector graphics and scaled to be readable.

22. In Figure 5G “++” label is missing from “Ca”.

23. Abbreviation usage is inconsistent. Figure 3 uses both “hours” and “hrs”.

24. Hash mark usage in plots is inconsistent. Figure 1A x-axis uses both log and linear scaling. Figure 3A axis has no marks but C does.

25. Check spelling and grammar in Figures. Figure 1D uses “Cell lyse” as a label. Also see the same problem in the graphic of Figure 4A.

26. Capitalization in nomenclature is inconsistent. Figure 4D uses “TAU” while panel C uses “Tau”.

27. Figure 3D uses “mTau” and “Mouse Tau” in the same image. Nomenclature should be consistent.

28. Figure 3D has size bars in the upper portions of the panels and not in the lower portions of the panel.

29. Figures denote the origin of Tau in images as human or mouse. In Figure 4F there is no indicator of whether Tau is human or mouse.

30. Symbol usage in plots is inconsistent. Figure 4B uses upright triangles for OND (III) and panel E uses upright triangles for OND (I). Other OND symbols are also flipped.

PLOS authors have the option to publish the peer review history of their article (what does this mean? ). If published, this will include your full peer review and any attached files.

**Do you want your identity to be public for this peer review?** For information about this choice, including consent withdrawal, please see our Privacy Policy .

Reviewer #1: No

Reviewer #2: No

Reviewer #3: No

**Figure resubmission:**
---

## [Decision Letter · Decision Letter 1]

Dear Dr. Safar,

We are pleased to inform you that your manuscript 'Structural exposure of different microtubule binding domains determines the propagation and toxicity of pathogenic tau conformers in Alzheimer’s disease.' has been provisionally accepted for publication in PLOS Pathogens.

Before your manuscript can be formally accepted you will need to complete some formatting changes, which you will receive in a follow up email. A member of our team will be in touch with a set of requests. Please also be sure to check the figures carefully and correct the minor formatting inconcistencies identified by Reviewer #3.

Best regards,

Suzette A. Priola

Academic Editor

PLOS Pathogens

Surachai Supattapone

Section Editor

PLOS Pathogens

Sumita Bhaduri-McIntosh

Editor-in-Chief

PLOS Pathogens

orcid.org/0000-0003-2946-9497

Michael Malim

Editor-in-Chief

PLOS Pathogens

orcid.org/0000-0002-7699-2064

Reviewer Comments (if any, and for reference):

Reviewer's Responses to Questions

**Part I - Summary**

Reviewer #1: The authors sufficiently addressed all of my concerns.

Reviewer #3: The authors put in considerable effort to take care of the various problems identify with the paper. The figures are in a much better spot and are more clear than the original version of the paper. The explanations provided clarify many of the problem points of the paper.

**Part II – Major Issues: Key Experiments Required for Acceptance**

Reviewer #1: (No Response)

Reviewer #3: -Supplemental Figure 6 B is still lacking a y-axis.

-The issue discussed with the size bars has not been addressed in figure 3 D. Either correct the size of the bars or place the size of each bar under each bar in each image.

-Some nomenclature in figures is inconsistent. For example hydroxyls are indicated with a dot before and after OH in the same figure.

**Part III – Minor Issues: Editorial and Data Presentation Modifications**

Reviewer #1: (No Response)

Reviewer #3: (No Response)

PLOS authors have the option to publish the peer review history of their article (what does this mean? ). If published, this will include your full peer review and any attached files.

**Do you want your identity to be public for this peer review?** For information about this choice, including consent withdrawal, please see our Privacy Policy .

Reviewer #1: No

Reviewer #3: No

---

## [Editor Report · Acceptance letter]

Dear Dr. Safar,

We are delighted to inform you that your manuscript, "Structural exposure of different microtubule binding domains determines the propagation and toxicity of pathogenic tau conformers in Alzheimer’s disease.," has been formally accepted for publication in PLOS Pathogens.

Best regards,

Sumita Bhaduri-McIntosh

Editor-in-Chief

PLOS Pathogens

orcid.org/0000-0003-2946-9497

Michael Malim

Editor-in-Chief

PLOS Pathogens

orcid.org/0000-0002-7699-2064